# High-Resolution Ecosystem Model of the Puck Bay (Southern Baltic Sea)—Hydrodynamic Component Evaluation

**Dawid Dybowski** [1,*], **Jaromir Jakacki** [2], **Maciej Janecki** [1], **Artur Nowicki** [1], **Daniel Rak** [3] and **Lidia Dzierzbicka-Glowacka** [1,*]

1   Physical Oceanography Department, Ecohydrodynamics Laboratory, Institute of Oceanology Polish Academy of Sciences, Powstańców Warszawy 55, 81-712 Sopot, Poland; mjanecki@iopan.pl (M.J.); anowicki@iopan.pl (A.N.)

2   Physical Oceanography Department, Ocean and Atmosphere Numerical Modeling Laboratory, Institute of Oceanology Polish Academy of Sciences, Powstańców Warszawy 55, 81-712 Sopot, Poland; jjakacki@iopan.pl

3   Physical Oceanography Department, Observational Oceanography Laboratory, Institute of Oceanology Polish Academy of Sciences, Powstańców Warszawy 55, 81-712 Sopot, Poland; rak@iopan.pl

*   Correspondence: ddybowski@iopan.pl (D.D.); dzierzb@iopan.pl (L.D.-G.);
    Tel.: +48-587-311-912 (D.D.); +48-587-311-915 (L.D.-G.)

**Abstract:** In recent years, thanks to the enormous computational power of modern supercomputers, modeling has become one of the most highly evolving scientific fields. It is now possible to describe relatively large physical bodies and to study the changes occurring in these bodies with resolution never attainable before. The paper describes the initial implementation of the EcoPuckBay model system and presents the results of the model simulations compared to observations from monitoring stations and other model reanalyses. High correlation between model results and observations has been confirmed both in terms of spatial and temporal approach. Data acquired via simulations of the EcoPuckBay model was deployed in the project archive database. The dedicated service was created, allowing the user to visualize all produced hydrodynamic parameters as raster maps, time series, and/or cross-sections. This functionality is available online via the official WaterPUCK project website in the services web section. In the next stage of the project, this service will be upgraded to an operational state and forecasts will be added.

**Keywords:** WaterPUCK; Puck Bay; 3D model; hydrodynamic variables

## 1. Introduction

The Puck Bay is part of the Gdańsk Basin (southern Baltic Sea). It is separated from deep-water areas by the Hel Peninsula. Puck Bay consists of the inner part called Puck Lagoon and the outer part of Puck Bay. The boundary between them runs from the Rybitwia Sandbank to the Cypel Rewski and has two straits within which there is an intensive water exchange between the Puck Lagoon and the outer part of the Puck Bay. The commonly accepted eastern border of the Puck Bay is the line connecting the Hel Peninsula with Kamienna Góra [1].

The Puck Bay is an example of a region that is highly vulnerable to anthropogenic impact. Therefore, it has been included into Natura 2000. As a result, it requires preservation or restoration of "favorable conservation status" of species and habitats by introducing appropriate "protection measures". The strategic actions and the policy of the authority of the Puck District regarding the environmental protection involve not only the respect of the Natura 2000 legislation but also realization

of European legislation including Water Framework Directive, Marine Strategy Framework Directive, Habitats Directive, Baltic Sea Action Plan and the strategic program of the environment protection for the Puck District. The main aims of the Puck District policy are improvement of the environment, sustainable development, protection from climate change, and protection of the natural resources such as water.

There are several models describing processes occurring in marine environment in use such as: HIROMB (High-Resolution Operational Model for the Baltic Sea), RCO (Rossby Centre Ocean Model), NEMO (Nucleus for European Modeling of the Ocean) and biogeochemical model SCOBI (Swedish Coastal and Ocean BIogeochemical model), HIRLAM (High-Resolution Limited Area Model) or ecosystem model of the Baltic Sea BALTSEM. At the end of 2015, the SatBałtyk System has been initiated in the IO PAN (Institute of Oceanology of the Polish Academy of Sciences). It enables efficient and systematic monitoring of the Baltic Sea Environment state based on innovative satellite techniques backed up with mathematical models of processes occurring in the sea. As part of the grant and continued work a three-dimensional model of the ecosystem 3D CEMBS (3D Coupled Ecosystem Model of the Baltic Sea) was improved and expanded. 3D CEMBS model (http://www.cembs.pl) generates 48-h forecast which include currents, temperature, salinity and ice parameters. In addition, the model forecasts ecological parameters i.e., nutrients, dissolved oxygen concentration and biomass of phytoplankton and zooplankton in the entire water column.

The EcoPuckBay model is being developed as part of the WaterPUCK project. The aim of the project is to create an integrated information and predictive service for the Puck District through the development of a computer system providing WaterPUCK service, which will clearly and practically assess the impact of farms and land-use structures on surface waters and groundwater in the Puck District, and consequently on the quality of the waters of the Puck Bay [2,3]. The construction of the service is based on in situ research, environmental data (chemical, physicochemical and hydrological) and numerical modeling. WaterPUCK service is an integrated system consisting of computer models interconnected with each other, operating continuously by supplying it with meteorological data and consists of 4 main modules:

- a comprehensive model of surface water runoff based on SWAT code,
- a numerical model of groundwater flow based on MODFLOW code,
- a three-dimensional numerical model of the Puck Bay ecosystem,
- a calculator of farms in Puck District as an interactive application [4].

## 2. Materials and Methods

The development of high-resolution numerical models that are able to simulate both physical and biogeochemical processes driving the state of the marine environment is time-consuming and requires broad knowledge of multiple events that occur in the modeled ecosystem. That requires access to information of direct and indirect forcing factors such as border fluxes, deposition from land and inputs that influence the dynamic of changes within the model domain. In order to match the complexity of this task and increase the efficiency of the development process, an existing state-of-the-art Community Earth System Model had been redesigned and adapted to the research area, allowing us to deliver an excellent tool for the purpose of either reanalysis or forecasting. We named it EcoPuckBay which corresponds to ecohydrodynamic model of the Puck Bay.

Here we describe an end-development (version 1.0) hydrodynamic configuration of the EcoPuckBay model. Biogeochemical configuration setup and results will be presented in the nearest future as a separate paper.

### 2.1. Study Area

The southern part of the Baltic Sea enclosing the Puck District is a tourist popular region that is also heavily influenced by an anthropogenic activity of local residents and farming. This makes the

Puck Bay a natural reservoir for waste deposition of fertilizers and other inputs delivered through soil, groundwater, river or direct deposition. In order to assess the possibility and scale of an eutrophication and water pollution the area of interest and effective domain in the EcoPuckBay model covers the western part of Gulf of Gdańsk (Figure 1). It can be divided further into a shallow part known as Puck Bay and the semi-enclosed Puck Lagoon to the west.

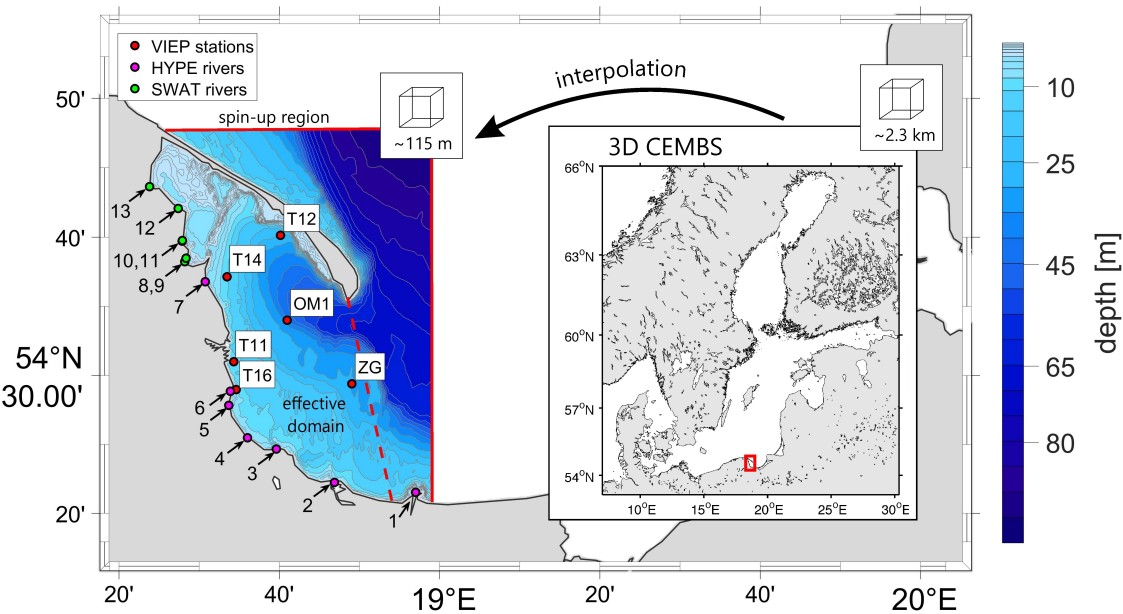

**Figure 1.** Model effective domain with topography, locations of sampling stations used for evaluation. and locations of mouths of watercourses included in domain. Green dots represent rivers that in the final product will be delivered using SWAT hydrological model.

This region is massively shaped and influenced by a several factors that can pressure the state of physical and biochemical parameters. One of the most important factor influencing region's unique ecosystem is topography. The average depth of the gulf is about 50 m, with the maximum depth (Gdańsk Deep) of 118 m. From the north-east it is surrounded by Hel Peninsula which serves as a natural barrier for mixing with the open waters of the Baltic Sea keeping the salinity ranging mostly within the range of 7–8 PSU with a deviation of around 1 PSU. Puck Bay is also heavily influenced by the river discharge from land resulting in lowered salinity especially in the coastal surface waters. The largest river in the region is Vistula, discharging an average of over 1000 $\text{m}^3 \text{ s}^{-1}$ of fresh water. Water temperature in the region ranges from over 20 °C at the surface during summer, with the maximum usually in August, to around 2 °C in February. Water stratification is frequent during the warmer months, leading to the occurrence of seasonal thermocline. During the winter seasons thermocline declines and water become well-mixed.

*2.2. EcoPuckBay Configuration*

EcoPuckBay origins from Community Earth System Model (CESM) coupled global climate model (http://www.cesm.ucar.edu/models/ccsm4.0) by NCAR. CESM is a state-of-the-are model system consisting of five separate components with an additional coupler controlling time, exciting forces, domains, grids and information exchange between the models. For the purpose of the WaterPUCK project, CESM was downscaled and adapted for the Puck Bay region for further development at the Institute of Oceanology, Polish Academy of Sciences.

EcoPuckBay's horizontal resolution is 1/960° which amounts to a nominal resolution of 115 m. The vertical resolution is 0.4–0.6 m in the upper layers down to 3 m, and then gradually increases to 5 m at depth, with a total of 24 layers (Table 1). The vertical discretization uses the *z* formulation and the bottom topography is based on Baltic Sea Bathymetric Database (BSBD) from the Baltic Sea

Hydrographic Commission [5]. The bathymetric data were interpolated into the model grid using Kriging method. The ocean model time step is 12 s.

**Table 1.** EcoPuckBay's vertical resolution.

| Model Level | Thickness (m) | Lower Depth (m) | Mid-Depth (m) |
|---|---|---|---|
| 1 | 0.60 | 0.60 | 0.30 |
| 2 | 0.40 | 1.00 | 0.80 |
| 3 | 0.40 | 1.40 | 1.20 |
| 4 | 0.40 | 1.80 | 1.60 |
| 5 | 0.40 | 2.20 | 2.00 |
| 6 | 0.50 | 2.70 | 2.45 |
| 7 | 0.60 | 3.30 | 3.00 |
| 8 | 0.80 | 4.10 | 3.70 |
| 9 | 1.00 | 5.10 | 4.60 |
| 10 | 1.40 | 6.50 | 5.80 |
| 11 | 1.80 | 8.30 | 7.40 |
| 12 | 2.50 | 10.80 | 9.55 |
| 13 | 4.00 | 14.80 | 12.80 |
| 14 | 5.00 | 19.80 | 17.30 |
| 15 | 5.00 | 24.80 | 22.30 |
| 16 | 5.00 | 29.80 | 27.30 |
| 17 | 5.00 | 34.80 | 32.30 |
| 18 | 5.00 | 39.80 | 37.30 |
| 19 | 5.00 | 44.80 | 42.30 |
| 20 | 5.00 | 49.80 | 47.30 |
| 21 | 5.00 | 54.80 | 52.30 |
| 22 | 5.00 | 59.80 | 57.30 |
| 23 | 5.00 | 64.80 | 62.30 |
| 24 | 5.00 | 69.80 | 67.30 |

The ocean component in EcoPuckBay is based on POP using three-dimensional equations of motion with hydrostatic and Boussinesq approximations. Main ocean equations in the spherical system are presented below.

Equations of horizontal motion:

$$\frac{\partial u}{\partial t} + L(u) - fv = -\frac{1}{\rho_0 a \cos \phi}\frac{\partial p}{\partial \lambda} + \frac{\partial}{\partial z}\left(K_M \frac{\partial u}{\partial z}\right) + B_M \nabla_H^4 u, \tag{1}$$

$$\frac{\partial v}{\partial t} + L(v) - fu = -\frac{1}{\rho_0 a}\frac{\partial p}{\partial \phi} + \frac{\partial}{\partial z}\left(K_M \frac{\partial v}{\partial z}\right) + B_M \nabla_H^4 v. \tag{2}$$

The momentum equation along the vertical direction within the hydrostatic approximation:

$$\frac{\partial p}{\partial z} = -\rho g. \tag{3}$$

Continuity equation:

$$\frac{1}{a \cos \phi}\frac{\partial u}{\partial \lambda} + \frac{1}{a \cos \phi}\frac{\partial (v \cos \phi)}{\partial \phi} + \frac{\partial w}{\partial z} = 0 \tag{4}$$

The equation of heat and salt transport:

$$\frac{\partial T}{\partial t} + L(T) = \frac{\partial}{\partial z}\left(K_D \frac{\partial T}{\partial z}\right) + B_D \nabla_H^4 T, \tag{5}$$

$$\frac{\partial S}{\partial t} + L(S) = \frac{\partial}{\partial z}\left(K_D \frac{\partial S}{\partial z}\right) + B_D \nabla_H^4 S, \tag{6}$$

The equation of state:

$$\rho = \rho(S, T, p). \tag{7}$$

where: $u$, $v$ horizontal components of velocity, $w$ vertical component of velocity, $g$ gravitational acceleration, $p$ pressure, $T$, $S$ temperature and salinity, $\rho_0$ average water density, $\lambda$, $\phi$ latitude and longitude, $a$ effective Earth radius, $t$ time, $f = 2\Omega \sin\phi$ the Coriolis parameter ($\Omega$ the rotation rate of the Earth), $L$ advection operator, $\nabla_H^4$ horizontal biharmonic operator, $K_M$ biharmonic vertical eddy viscosity, $B_D$ biharmonic horizontal eddy viscosity. Horizontal coefficients $B_D$, $B_M$ are assumed as constants. Vertical mixing in EcoPuckBay is determined by k-profile parameterization (KPP) [6] determining the vertical coefficients $K_D$, $K_M$. In the current setup, an enhancement of the KPP Scheme with Bottom Boundary Layer introduced by Durski [7] has been applied:

$$C_d = \kappa^2 \left( \ln \frac{dz}{z_r} \right)^{-2}, \tag{8}$$

where $C_d$ is drag coefficient $\kappa$ is von Karman's constant, $dz$ is the distance from the bottom to grid point, $z_r$ is a roughness height specified as 0.5 cm (value tuned up based on the water flow through The Sund in 3D CEMBS model). Parametrization of the MWJF state equation, developed by McDougall et al. [8], was applied in the model. The EcoPuckBay chosen parametrization is very similar to the 3D CEMBS model. The model uses preconditioned conjugate gradient solver (PCG) in the barotropic part. Our configuration uses Lax–Wendroff with 1-D flux limiters advection scheme. We also introduced biharmonic (bilaplacian operator) horizontal friction (momentum) as well as biharmonic horizontal mixing for tracers. POP is a model with free surface, and vertical velocity on the surface amounts to:

$$w = \frac{\partial \eta}{\partial z}, \; z = 0, \tag{9}$$

where: $\eta$ free surface elevation. Stress at the bottom $\tau_b$ is expressed as:

$$\frac{\tau_b}{\rho} = \frac{C_d \vec{U} \left| \vec{U} \right|}{d^2}, \tag{10}$$

where: $d$ thickness of the benthic layer, $C_d$ drag coefficient, $\vec{U}$ water velocity vector at the bottom. Ocean in EcoPuckBay is coupled to the sea-ice model based on Community Ice CodE (CICE). The main equation solved by CICE is:

$$\frac{\partial k}{\partial t} = -\nabla \cdot (k\boldsymbol{U}) - \frac{\partial}{\partial h}(ck) + \psi, \tag{11}$$

where: $\boldsymbol{U}$ is horizontal ice velocity, $c$ the ice growth rate, $\psi$ ridging redistribution, $k$ distribution of ice thickness.

*2.3. Open Boundary*

EcoPuckBay model results discussed in this paper are limited to an effective area of Puck District surroundings. Whole grid of the model however, covers wider domain reaching to Gdańsk Deep latitudes. This is made to ensure that boundary conditions are simulated properly. Along the line of the EcoPuckBay's northern border an information exchange with 2.3 km prognostic model 3D CEMBS takes place (see Figure 1). Results from 3D CEMBS serve as forcing fields to EcoPuckBay through a sequential information exchange. The algorithm performing the connection ensures mass and energy conservation. Currently exists two approaches that can include tides in the ocean model. First is adding tidal potential in the barotropic equation. The second provides tides in the lateral boundaries (it can be done as a surface variation or Flather lateral boundaries). The first approach does not work properly for small seas. The main reason is that the water mass is small, thus the result of such implementation provide proper period and phase of the tidal constituents, but the amplitude

is usually too low. The second approach is indirectly included in the EcoPuckBay model because lateral boundaries are included in the coarse resolution model in an open boundary area via sea level provided by Goteborg station. Thus, although there is a tacit assumption that the Baltic Sea has no tides, they are being broadcasted via lateral boundaries.

### 2.4. Atmosphere Forcing

EcoPuckBay model is forced by 48-h meteorological forecasts from the UM model delivered by the Interdisciplinary Centre for Mathematical and Computational Modeling of Warsaw University (ICM UW). Following external fields are used:

- 2 m air temperature and specific humidity,
- sea level pressure,
- precipitation,
- short and long wave radiation,
- 10 m wind speed,
- air density.

### 2.5. River Discharge

At this stage of the development the volume data of river discharge come from the Hydrological Predictions for the Environment (HYPE) model. It is a semi-distributed, physically based catchment model, which simulates water flow and substances on their way from precipitation through different storage compartments and fluxes to the sea [9–11]. We used historical time series from the period 1980–2010 for the Europe geographical domain available as a daily means. The spatial resolution is given by landscape delineation into catchments, for which HYPE data represents average conditions or the outlets. Six HYPE catchments have been taken into account to resolve 13 discharge locations (rivers, canals and streams) of watercourses with their estuaries running alongside the EcoPuckBay model domain (Table 2). Volumes for the years past 2010 have been calculated as a long-term means from the available 30-year period.

**Table 2.** Locations of mouths of watercourses included in the EcoPuckBay model domain.

|  | Watercourse | Longitude | Latitude |
|---|---|---|---|
| 1 | Vistula | 18.95 | 54.35 |
| 2 | Bold Vistula | 18.78 | 54.37 |
| 3 | Still Vistula | 18.66 | 54.41 |
| 4 | Oliwski Stream | 18.60 | 54.42 |
| 5 | Kamienny Stream | 18.56 | 54.46 |
| 6 | Kacza | 18.56 | 54.48 |
| 7 | Ściekowy Canal | 18.51 | 54.61 |
| 8 | Zagórska Stream | 18.47 | 54.63 |
| 9 | Reda | 18.47 | 54.64 |
| 10 | Mrzezino Canal | 18.46 | 54.66 |
| 11 | Gizdepka | 18.46 | 54.66 |
| 12 | Żelistrzewo Canal | 18.45 | 54.70 |
| 13 | Płutnica | 18.39 | 54.72 |

At the final stage of the development, the HYPE information about the water volume discharged by rivers with their mouths located within the Puck District (Figure 1, numbers from 8 to 13) will be replaced by the hydrological model SWAT that is being implemented as one of the WaterPUCK project's stages [12–16]. SWAT model includes the preparation of innovative and complex hydrological model including meteorological data (precipitation, wind, temperature, atmospheric pressure). The proposed solution is based upon real time observation (local weather station) and on short-term weather forecasts (the ICM UW web page). The hydrological computations will be performed with SWAT software.

The transformation of precipitation data into surface runoff will be achieved with the SCS (Soil Conservation Service) Curve Number procedure through the accumulated runoff volume and the time of concentration (the time from the beginning of a rainfall event until the entire sub-basin area is contributing to flow at the outlet).

## 2.6. Simulation Run

To calculate hydrodynamic conditions of the Puck Bay area a simulation run has been performed for the time period January 2011 to December 2018. Since we had no access to reliable long-term and spatially representative in situ database preceding year 2014 that could be used to make a comparison with model results, we threat the starting three years as a spin-up stage, even though it would be enough to use a simulation with a spin-up period of one year or less on a homogeneous domain of this size and resolution.

There is no data assimilation module in the current development version of the model yet; however, it will be introduced in the next stages of the project's workflow.

## 2.7. Datasets Used for Evaluation

We used several sources of in situ samples to evaluate EcoPuckBay model. That includes measurements taken throughout the monitoring activities of the Voivodship Inspector of Environmental Protection in Gdańsk and vertical profiles of temperature and salinity recorded in 2018 during one of the measurement campaigns of the s/y Oceania along the Southern Baltic coast. In addition, the numerical data of physical parameters calculated with the circulation model system NEMO-Nordic that comes from the Marine Copernicus database was processed too. A brief summary description of each of the databases used in this paper for the evaluation purpose has been presented below.

### 2.7.1. VIEP

The biggest set of the in situ data used for the EcoPuckBay model evaluation performed in this paper are samples collected during regular measurements of the marine environment of the Baltic Sea conducted by Voivodship Inspectorate of Environmental Protection (VIEP) in Gdańsk. We used data from 6 stations located within the model domain (Figure 1) taken between January 2014 and December 2016. VIEP's scope, methods of monitoring and the manner of assessing the status of the Baltic Sea has been defined by the provisions of the Water Law Act and the monitoring program of marine waters adopted by the Council of Ministers, which implements the requirements of Article 11 of Directive of the European Parliament and of the Council 2008/56/EC of 17 June 2008, establishing a framework for community action in the field of marine environmental policy. Monitoring service performed by VIEP constitute the fulfilment of Poland's obligations arising from the Convention on the protection of the marine environment of the Baltic Sea area, and since 15 July 2014, of Article 11 of Directive 2008/56/EC of the European Parliament and establishing a framework for community action in the field of marine environmental policy.

### 2.7.2. s/y Oceania

Hydrodynamical data samples analyzed in this paper were collected during the summer 2018 regular cruise of s/y Oceania on the Puck Bay. (data not yet published). Vertical sections for 18 stations (Figure 2) were performed using a Seabird 49 Conductivity Temperature Depth (CTD) probe.

CTD sensor's accuracy was 0.0003 mS cm$^{-1}$ for Conductivity, 0.002 °C for Temperature and 0.1% for the pressure. Temperature and conductivity sensors of CTD system were calibrated annually, post-cruise, by the manufacturers. The vertical sampling was higher than model resolution (from 10–40 readings per model level, depending on its depth), therefore ship readings were averaged to match EcoPuckBay resolution.

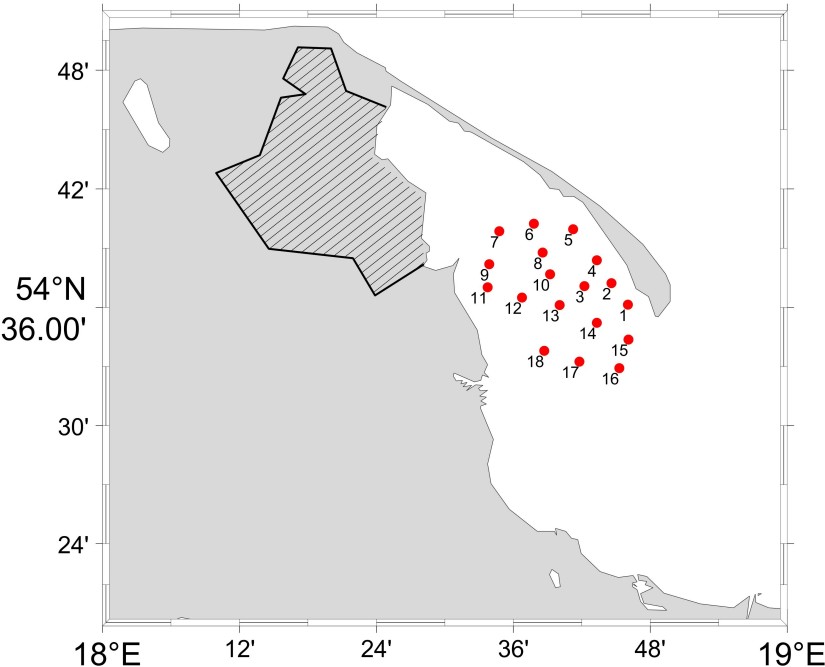

**Figure 2.** Location of stations for vertical profiles collected from s/y Oceania.

### 2.7.3. NEMO-Nordic

The source of the numerical results used in the manuscript is the data downloaded from Marine Copernicus (http://marine.copernicus.eu) database. This product origin from a Baltic Sea physical reanalysis for the period January 1993 to December 2016 at 3.9 km horizontal resolution and had been produced using the ice-ocean model NEMO-Nordic. Temperature, salinity for surface and bottom layer and currents data were delivered as daily means.

The circulation model NEMO-Nordic is based on NEMO-3.6 model version. It has been the operational ocean and sea-ice forecasting model used at SMHI (the Swedish Meteorological and Hydrological Institute) since 2016 [17,18]. NEMO-Nordic domain covers whole Baltic Sea extended to the North Sea area.

## 3. Results

### 3.1. Model Evaluation

To evaluate the quality of the results obtained from the EcoPuckBay model for the period of January 2014 to December 2018, a set of basic statistical measures were calculated, such as Pearson correlation coefficient ($r$), root-mean-square-error (RMSE), standard deviation (STD) and bias (in terms of means).

The results of comparisons are presented in Sections 3.1.1 and 3.1.2 in graphical and tabular form. The obtained statistical parameters allowed verification of the model in terms of seasonal and spatial variability of simulated water temperature and salinity, which are the most essential physical parameters when describing the state of the marine environment.

### 3.1.1. Temperature

The quality of the temperature is assessed by comparing the modeled temperature with all available observations from VIEP and s/y Oceania stations. Statistical comparison is presented in form of Taylor diagrams [19] in Figure 3. To present result from time series and vertical profiles on one diagram all the statistics were normalized with standard deviation of reference measurement. Non-normalized values are presented in Table 3. In Figure 4 we present the average vertical water temperature profiles for all stations (Figure 4a) and for one selected station (Figure 4b).

**Table 3.** Statistical comparison between modeled temperature and reference data from in situ measurements (VIEP and s/y Oceania) and numerical data (NEMO-Nordic).

| Reference Data | Pearson's *r* | RMSE (°C) | STD (°C) | Bias (°C) |
|---|---|---|---|---|
| Time series (VIEP) | 0.97 | 1.45 | 5.67 | −0.83 |
| Time series (NEMO-Nordic) | 0.98 | 1.33 | 6.01 | −0.31 |
| Vertical profiles (s/y Oceania) | 0.92 | 2.85 | 5.29 | −1.16 |

Time series for separate VIEP stations can be found in the Appendix A (Figures A1–A6) as well as vertical profiles for separate s/y Oceania stations (Figure A7). The statistics calculated for each station has been summarized in Tables 4 and 5, for VIEP and s/y Oceania stations respectively.

The Pearson correlation coefficient *r* for all measured values is equal to 0.97 for VIEP stations and 0.92 for s/y Oceania stations. The RMSE from all VIEP stations is equal to 1.45 °C and 2.85 °C from all s/y Oceania stations.

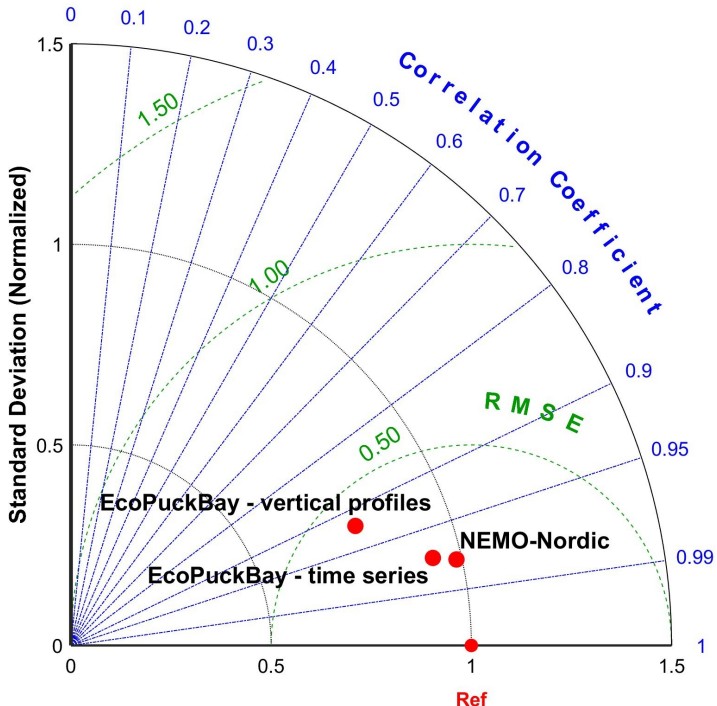

**Figure 3.** Taylor diagram for temperature.

The Pearson correlation coefficient calculated for the VIEP time series has the minimum value of 0.94 and the maximum value of 0.99. Such results indicate a very good representation of the variability characteristics of the modeled data with the measurement data. RMSE for T11, T14, T16 and OM1 stations does not exceed 1.5 °C, while on the other two stations it is around 2 °C. A negative bias value indicates that the modeled data for the considered stations have a lower value than the measurement data.

Analyzing the vertical temperature profiles from s/y Oceania, a high correlation between the EcoPuckBay model and the in situ measurements can be observed. At all measurement stations the Pearson correlation coefficient between modeled data and measured values is higher than 0.7. For stations from 12 to 17 the correlation exceeds 0.98, which indicates a perfect correspondence between the profiles and the correct implementation of the vertical mixing in the EcoPuckBay model. The lowest correlations occurred at stations 7, 9, 11 and, as you can see in Figure 2, these are the stations most far to the west, i.e., those on the shallow water and additionally strongly influenced by external forces, e.g., rivers.

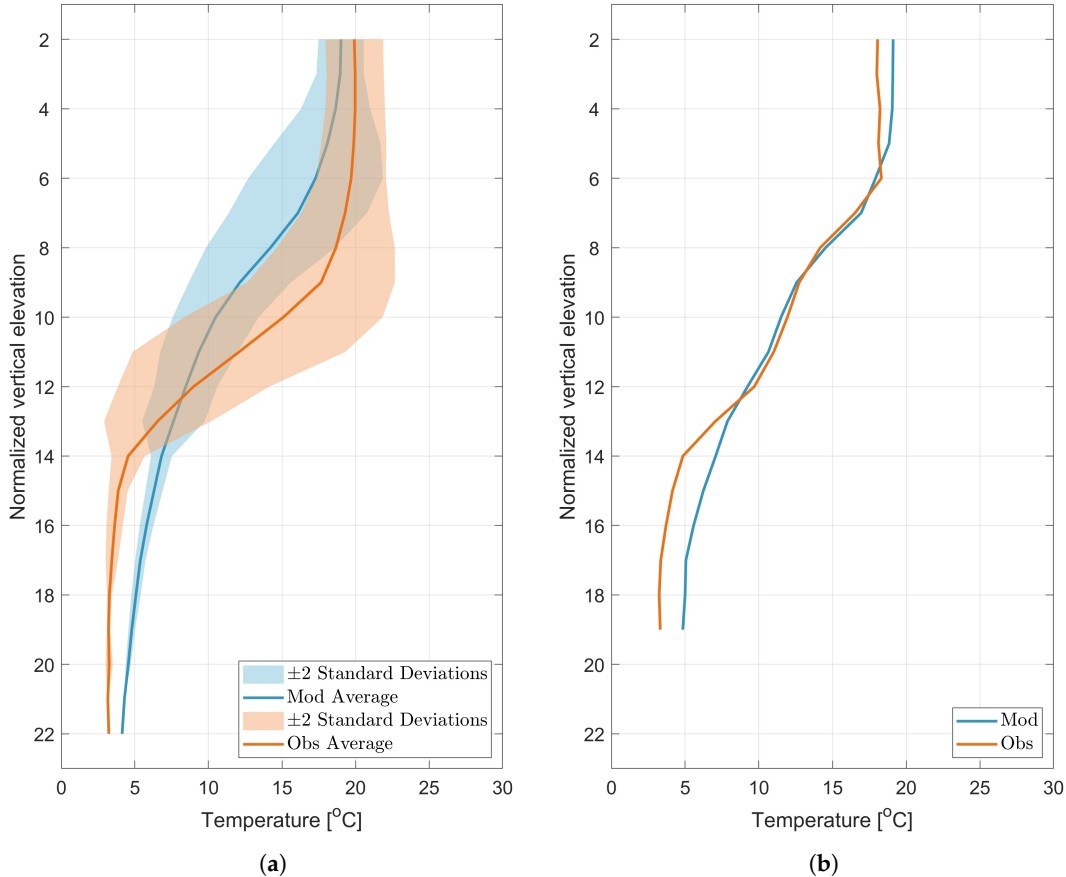

**Figure 4.** Temperature vertical profiles (**a**) from all stations (**b**) at station 13.

**Table 4.** Statistical comparison between model results and in situ measured water temperature at the surface level (VIEP).

| Station | Pearson's *r* | RMSE (°C) | STD (°C) | Bias (°C) |
|---------|---------------|-----------|----------|-----------|
| T11 | 0.99 | 1.24 | 5.48 | −0.99 |
| T12 | 0.94 | 2.08 | 5.37 | −1.12 |
| T14 | 0.98 | 1.41 | 5.45 | −1.24 |
| OM1 | 0.98 | 1.16 | 5.30 | −1.32 |
| T16 | 0.99 | 1.17 | 5.47 | −0.66 |
| ZG | 0.95 | 1.98 | 4.87 | −1.71 |

The STD at respective stations varies from the minimum value of 4.05 °C at station 2 to 5.64 °C at station 16. This is the expected result, especially for the day when the cruise took place because at virtually every station you can see a scratching thermocline which is the result of a weak mixing in this period and the formation of stagnant layers of water near the surface with a high temperature reaching from 18 to even more than 20 °C (depending on the profile station) which at the bottom drops to about 10 °C in places where there is a shallow to less than 5 °C for the deepest points in which the ship measured the profile. High STD reproduces this large variation from the surface to the bottom.

A negative bias on 15 of 18 profiles indicates that in the transition section of the thermocline, the model tends to lower the results, but at the bottom it reaches values close to those measured.

Generally, the model reproduces well the temperature in the water column, as demonstrated by high correlations and low RMSE which is formed at a level of about 3 °C.

**Table 5.** Statistical comparison between model results and in situ measured water temperature in vertical profiles (s/y Oceania).

| Station | Pearson's *r* | RMSE (°C) | STD (°C) | Bias (°C) |
|---|---|---|---|---|
| 1 | 0.95 | 3.44 | 4.68 | −0.77 |
| 2 | 0.84 | 4.23 | 4.05 | −1.96 |
| 3 | 0.95 | 2.87 | 5.18 | −1.24 |
| 4 | 0.92 | 3.58 | 4.30 | −1.27 |
| 5 | 0.95 | 2.31 | 4.59 | −0.45 |
| 6 | 0.91 | 2.86 | 4.88 | −1.67 |
| 7 | 0.72 | 3.71 | 5.03 | −3.19 |
| 8 | 0.93 | 2.70 | 5.61 | −0.71 |
| 9 | 0.77 | 2.74 | 4.18 | −3.19 |
| 10 | 0.95 | 2.79 | 5.61 | −0.71 |
| 11 | 0.77 | 2.66 | 4.13 | −3.88 |
| 12 | 0.99 | 1.06 | 5.23 | 0.24 |
| 13 | 0.99 | 0.90 | 5.46 | 0.81 |
| 14 | 0.98 | 1.66 | 5.41 | −0.04 |
| 15 | 0.98 | 2.15 | 5.21 | −0.34 |
| 16 | 0.98 | 1.60 | 5.64 | −0.38 |
| 17 | 0.98 | 1.41 | 5.47 | 0.22 |
| 18 | 0.93 | 2.82 | 5.16 | −1.75 |

### 3.1.2. Salinity

The quality of the salinity is assessed by comparing the modeled salinity with all available observations from VIEP and s/y Oceania stations. Statistical comparison is presented in form of Taylor diagrams in Figure 5. To present result from time series and vertical profiles on one diagram all the statistics were normalized with standard deviation of reference measurement. Non-normalized values are presented in Table 6. In the Figure 6 we present vertical water temperature profiles for all stations (Figure 6a) and for one selected station (Figure 6b).

**Table 6.** Statistical comparison between modeled salinity and reference data from in situ measurements (VIEP and s/y Oceania) and numerical data (NEMO-Nordic).

| Reference Data | Pearson's *r* | RMSE [PSU] | STD [PSU] | Bias [PSU] |
|---|---|---|---|---|
| time series (VIEP) | 0.58 | 0.67 | 0.60 | 0.16 |
| time series (NEMO-Nordic) | 0.17 | 0.97 | 0.70 | −0.24 |
| vertical profiles (s/y Oceania) | 0.90 | 0.40 | 0.84 | −0.03 |

Time series for separate VIEP stations can be found in the Appendix A (Figures A10–A15) as well as vertical profiles for separate s/y Oceania stations (Figure A16). The statistics calculated for each station has been summarized in Tables 7 and 8, for VIEP and s/y Oceania stations respectively.

The correlation coefficient *r* for all measured values is equal to 0.58 for VIEP stations and 0.90 for s/y Oceania stations. Therefore, it varies from 0.28 to 0.86 for VIEP and from 0.78 to 1.00 for s/y Oceania stations.

Pearson's correlation coefficients for time series from the VIEP salinity measurements ranges from 0.28 to 0.86. Station T11, for which the correlation coefficient reaches the lowest value, is located close to the Port of Gdynia, which may be the cause of such a result. RMSE does not exceed 1 PSU for all stations and is less than 0.5 PSU for half of them. The STD changes from 0.34 to 0.81 PSU depending on the station. The lowest time variability is observed in the T12 and T14 stations that are located nearest to Puck Lagoon. The bias is less than 0.3 PSU and always positive, i.e., the salinity calculated by the model is slightly higher than the salinity measured by the VIEP. This situation can be improved by attaching SWAT model data to the EcoPuckBay model.

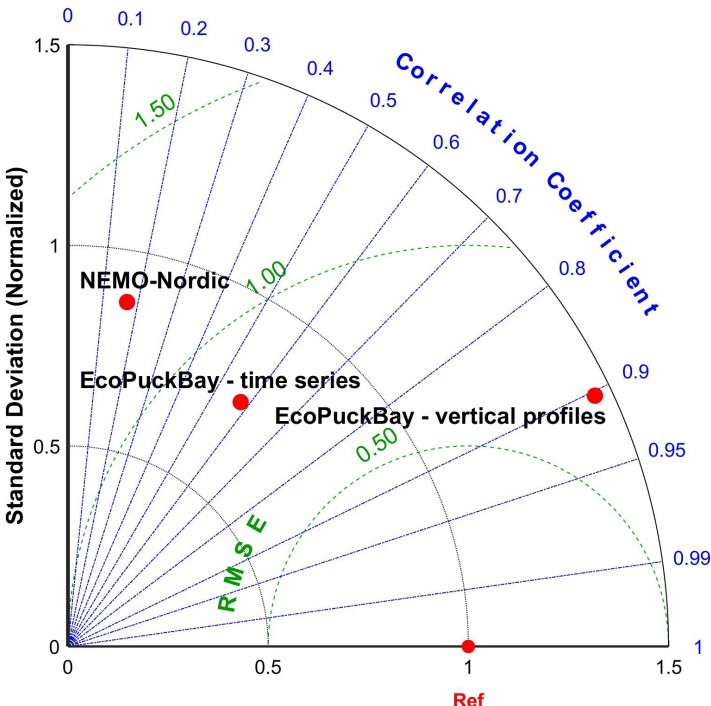

**Figure 5.** Taylor diagram for salinity.

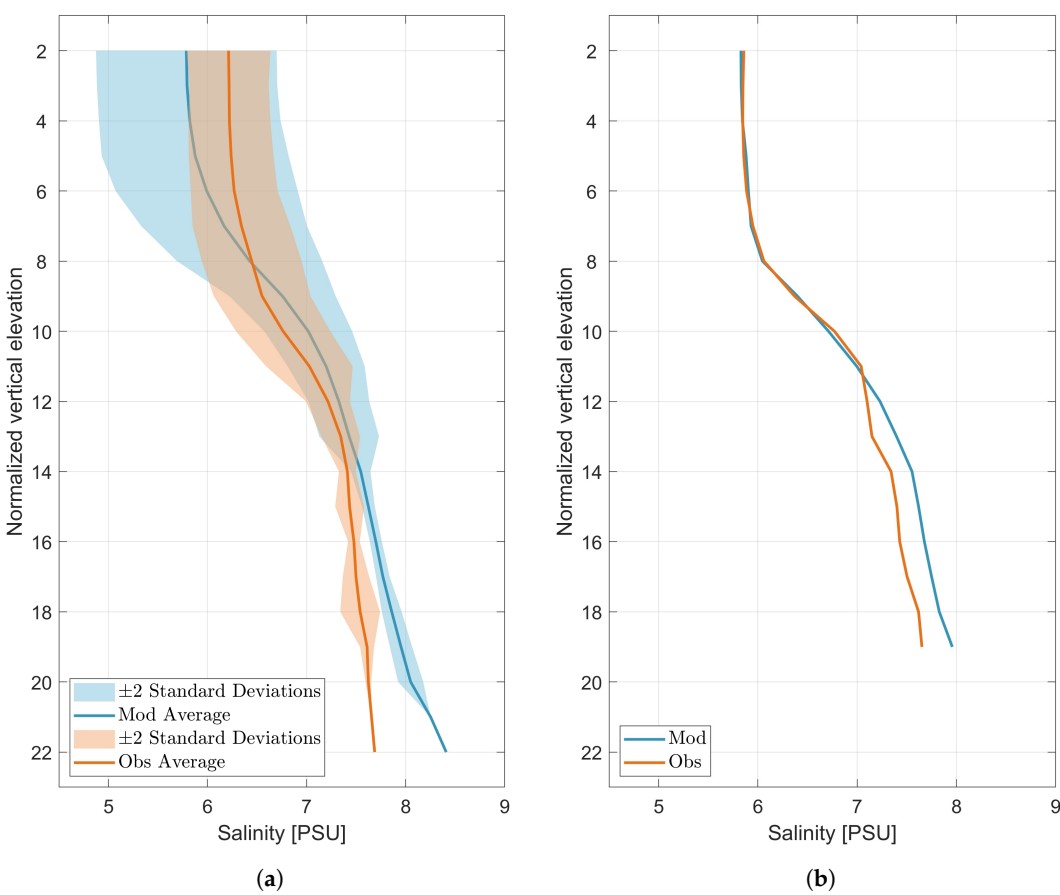

**Figure 6.** Salinity vertical profiles (**a**) from all stations (**b**) at station 16.

Comparing the vertical salinity profiles from the modeled data with the profiles from s/y Oceania measurements, we observe Pearson's correlation coefficients from 0.78 to 1.00. As with the temperature,

lower correlation coefficients occur at stations 7, 9, 11, which are located near the Rybitwia Sandbank which separates Puck Lagoon from the outer part of the Puck Bay. RMSE does not exceed 0.5 PSU except for the three mentioned stations and station 13.

**Table 7.** Statistical comparison between model results and in situ measured water salinity at the surface level (VIEP).

| Station | Pearson's $r$ | RMSE [PSU] | STD [PSU] | Bias [PSU] |
|---------|---------------|------------|-----------|------------|
| T11 | 0.28 | 0.99 | 0.54 | 0.21 |
| T12 | 0.53 | 0.43 | 0.37 | 0.12 |
| T14 | 0.64 | 0.35 | 0.34 | 0.19 |
| OM1 | 0.86 | 0.32 | 0.51 | 0.27 |
| T16 | 0.51 | 0.81 | 0.61 | 0.04 |
| ZG | 0.62 | 0.78 | 0.81 | 0.14 |

### 3.2. Simulation Results

Below we present spatial distributions of selected physical parameters in the Puck Bay region simulated by the EcoPuckBay model. To avoid exceeding paper size, we limited this section only to the monthly averages of water temperature, salinity, sea level and currents from the 5-year period of 2014–2018. Figures and Tables for the separate years were moved to the Appendix A (Figures A8–A9, A17–A22 and Tables A1–A15).

**Table 8.** Statistical comparison between model results and in situ measured water salinity in vertical profiles (s/y Oceania).

| Station | Pearson's $r$ | RMSE [PSU] | STD [PSU] | Bias [PSU] |
|---------|---------------|------------|-----------|------------|
| 1 | 0.97 | 0.14 | 0.58 | 0.21 |
| 2 | 0.91 | 0.20 | 0.47 | 0.24 |
| 3 | 0.99 | 0.17 | 0.72 | 0.10 |
| 4 | 0.96 | 0.14 | 0.49 | 0.18 |
| 5 | 0.98 | 0.19 | 0.53 | 0.01 |
| 6 | 0.92 | 0.28 | 0.58 | 0.04 |
| 7 | 0.78 | 0.48 | 0.72 | 0.06 |
| 8 | 0.95 | 0.32 | 0.77 | 0.00 |
| 9 | 0.80 | 0.69 | 0.92 | −0.41 |
| 10 | 0.96 | 0.31 | 0.82 | 0.00 |
| 11 | 0.88 | 0.64 | 0.92 | −0.36 |
| 12 | 0.96 | 0.49 | 0.93 | −0.42 |
| 13 | 0.98 | 0.60 | 0.99 | −0.37 |
| 14 | 0.99 | 0.32 | 0.79 | −0.01 |
| 15 | 0.98 | 0.21 | 0.76 | 0.19 |
| 16 | 1.00 | 0.12 | 0.81 | 0.10 |
| 17 | 0.98 | 0.28 | 0.86 | −0.10 |
| 18 | 0.95 | 0.39 | 0.95 | −0.09 |

### 3.2.1. Temperature

The average surface temperature inside domain was 10.3 °C. The lowest monthly mean temperature for specified model cell was in January 2016 with a value of −0.28 °C and the highest in July 2014 with a maximum exceeding 24 °C both inside Puck Lagoon (Figures A8–A9 and Table A1). The standard deviation from the average for whole period from 2014 to 2018 is nearly 6 °C, which indicates high seasonal variability of temperature in the Puck Bay presented in Figure 7 (based on Tables A1–A4).

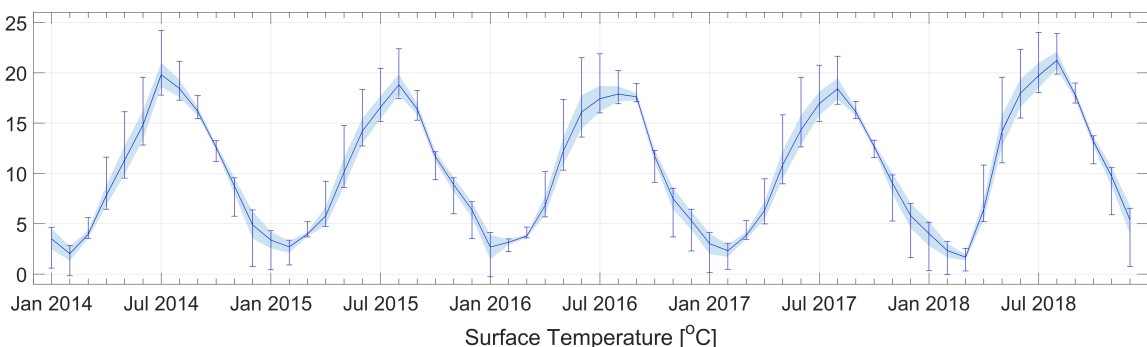

**Figure 7.** Monthly means of temperature at sea surface level for whole domain. Error bars represent minimum and maximum values. Solid line connects monthly means and shadow area shows standard deviation from the mean values.

Figure 8 shows the average surface temperature for the individual months of the 2014–2018 period. Puck Lagoon is characterized by the highest dynamics of temperature variability due to its geomorphological separation from the rest of the Puck Bay. In summer, the water in Puck Lagoon warms up to higher temperatures, while in winter it reaches lower temperatures than the rest of the Puck Bay (Figures A8 and A9). The highest average monthly water temperature in the surface layer for the whole domain was higher than 21 °C in August 2018. Moreover, from May to August 2018 the average monthly surface temperature for the whole domain was over 1.5 °C higher than the 5-year average for the whole domain (for the years 2014–2018) and for May, June and August, more than 2.2 °C from that average (Table A1). Additionally, in March 2018, monthly average surface temperature was the lowest for the entire modeled period and deviated from the average by nearly 1.8 °C.

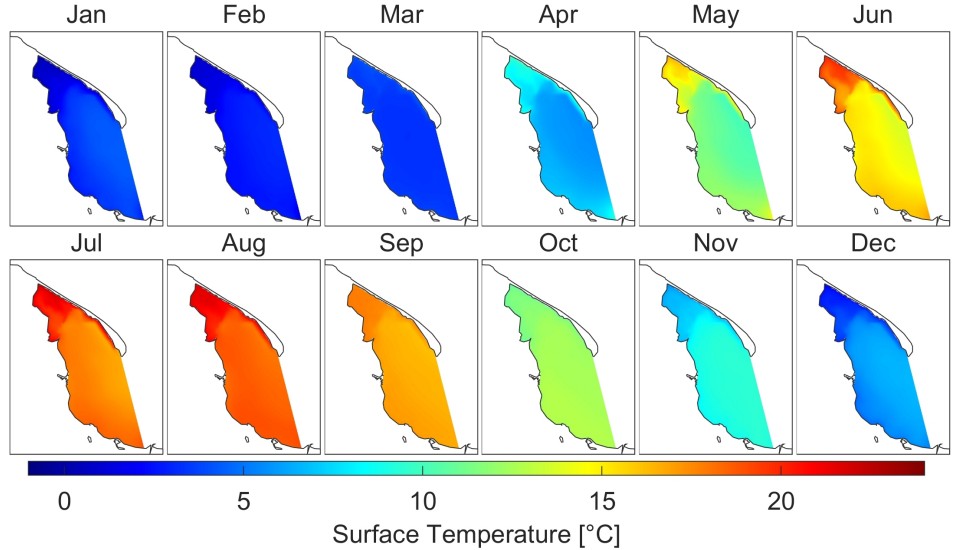

**Figure 8.** Monthly means of water temperature at sea surface level.

### 3.2.2. Salinity

The average surface salinity in the domain was 7.05 PSU. The lowest salinity value was calculated near the river mouths with a minimum of 0.18 PSU (March 2014) and the highest value on the eastern border of the region with a maximum of 8.19 PSU (January 2016). The standard deviation for whole modeled period was equal to 0.81 PSU. Figure 9 (based on Tables A5–A8) shows the seasonal variability of the monthly mean surface salinity inside entire domain. Monthly average salinity in the surface layer varies from 5.5 to about 8 PSU. For nearly 50% of the modeled months (28 out of 60) the standard deviation does not exceed 0.5 PSU. The most significant impact on the water salinity in the Puck Bay is the amount of freshwater inflow from rivers (mainly the Vistula river).

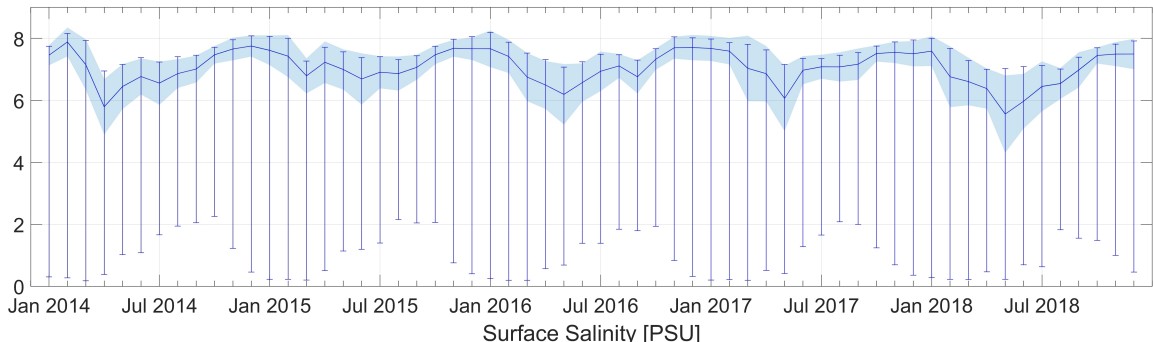

**Figure 9.** Monthly means of salinity at sea surface level for whole domain. Error bars represent minimum and maximum values. Solid line connects monthly means and shadow area shows standard deviation from the mean values.

Figure 10 presents the monthly averages of surface salinity from years 2014–2018. The lowest values of surface salinity were calculated in the vicinity of river mouths. The biggest influence on salinity of the Puck Bay has the Vistula river. The strong influence of fresh water between March and September is particularly pronounced (Figures A17 and A18). Analyzing the 5-year period (2014–2018), the lowest average salinity in the surface layer of the domain was in May, while the highest in December. From October to January, for all modeled years, the absolute value of the difference between the 5-year average and the monthly average for a given year did not exceed 0.15 PSU (Table A5).

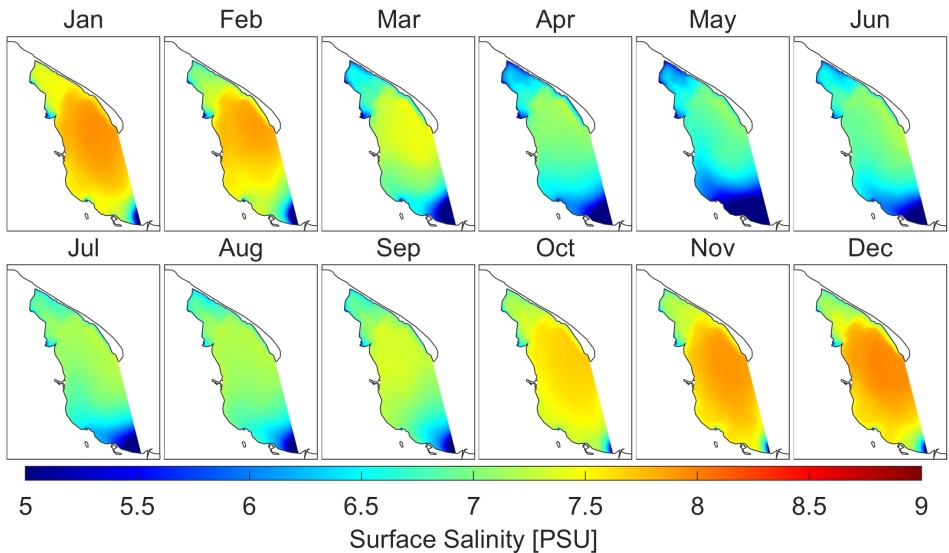

**Figure 10.** Monthly means of salinity at sea surface level.

### 3.2.3. Sea Surface Height

The lowest monthly average sea level calculated in the model was -3.35 cm in December 2016 inside Puck Lagoon. The highest monthly average sea level was 2.81 cm in November 2014 also inside Puck Lagoon (Figure 11 based on Tables A9–A12).

The monthly average sea surface height oscillates between a minimum value of about $-3.4$ cm and a maximum value of 2.8 cm with mean value of $-0.18$ cm for whole domain. Standard deviation from the mean is equal to 0.6 cm which indicates a small variation in the average sea surface height in the Puck Bay (see Figure 12). As expected, extreme values of sea surface height occurs mainly along the coasts because the monthly average sea surface height is determined by long-term wind forcing (Figures A19 and A20).

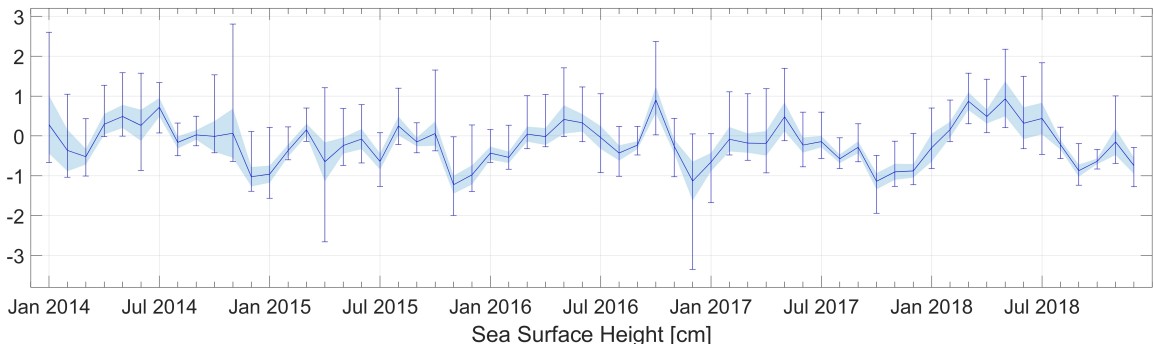

**Figure 11.** Monthly means of sea surface height for whole domain. Error bars represent minimum and maximum values. Solid line connects monthly means and shadow area shows standard deviation from the mean values.

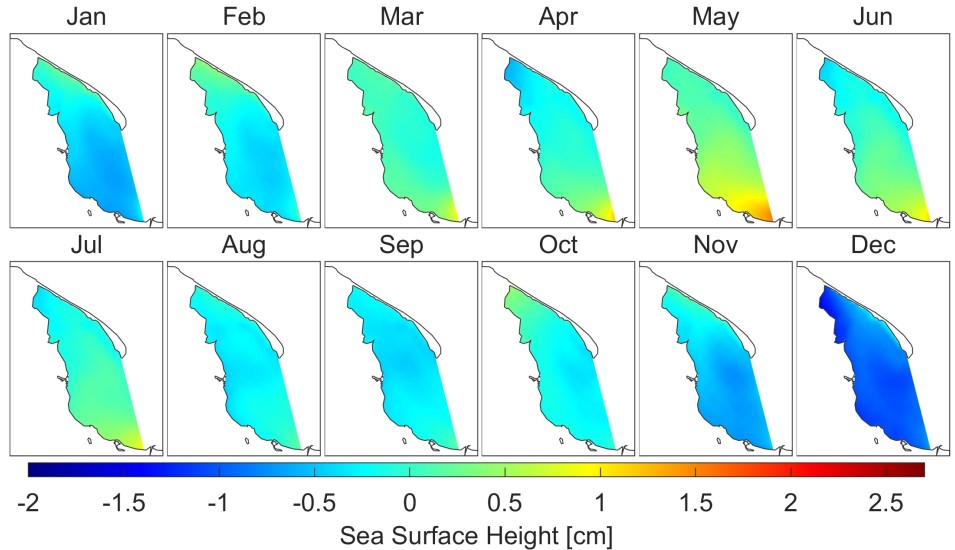

**Figure 12.** Monthly means of sea surface height.

### 3.2.4. Currents

The average value of the current for the whole domain in the whole period was equal to 3.36 cm·s$^{-1}$ with the standard deviation from the average equal to 2.34 cm·s$^{-1}$ (Figure 13 based on Tables A13–A15). The maximum value of the monthly average horizontal velocity was above 27 cm·s$^{-1}$ and was calculated on July 2018 near Hel.

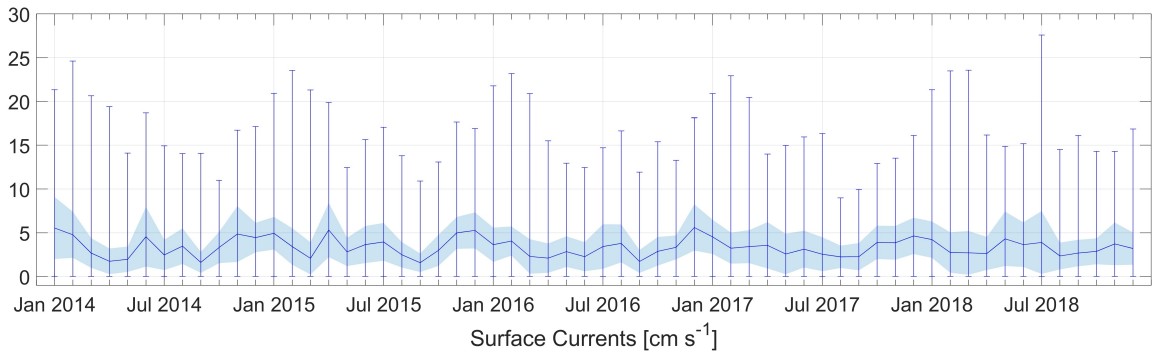

**Figure 13.** Monthly means of currents at sea surface level for whole domain. Error bars represent minimum and maximum values. Solid line connects monthly means and shadow area shows standard deviation from the mean values.

Figure 14 shows the monthly average spatial distribution of currents over the whole modeled period. In the monthly average, the picture of circulation inside the Puck Bay does not show a vortex character. On the other hand, Figure 15a shows an exemplary image of anticyclonic eddy produced inside the Puck Bay with a temperature cross-section (Figure 15b). According to numerical simulations carried out for the period 1960–1969 by Osiński [20], the vortex size in this area is about 15 km and the average duration about 7 days. Based on our calculations using the EcoPuckBay model developed for WaterPUCK service the diameter of the anticyclonic eddy is about 15–20 km and its duration time is between 7–10 days; in the case shown in Figure 15 it was 10 days (from 21 May to 30 May). Velocities on the east side of the vortex reached 50 cm·s$^{-1}$, on the west side up to 40 cm·s$^{-1}$ and on the inside up to 20 cm·s$^{-1}$.

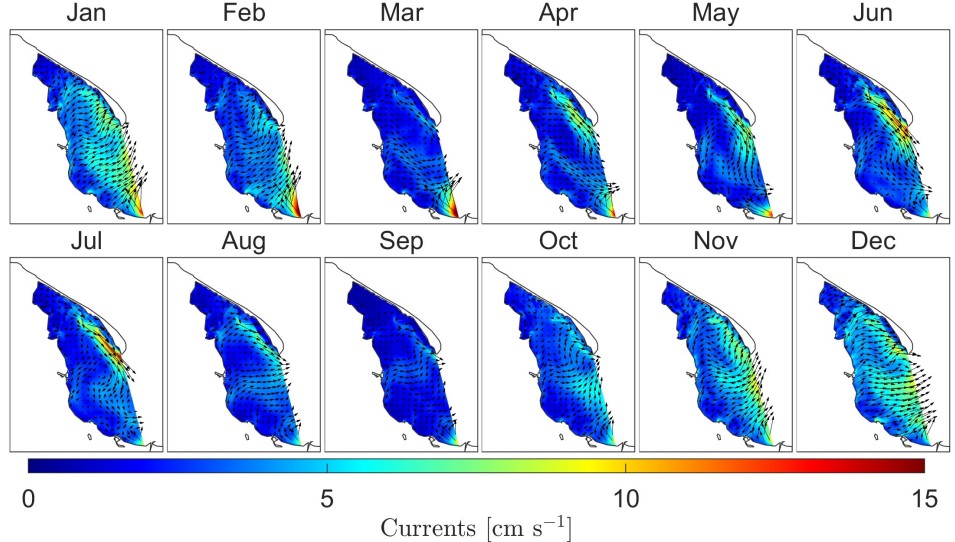

**Figure 14.** Monthly means of currents at sea surface level.

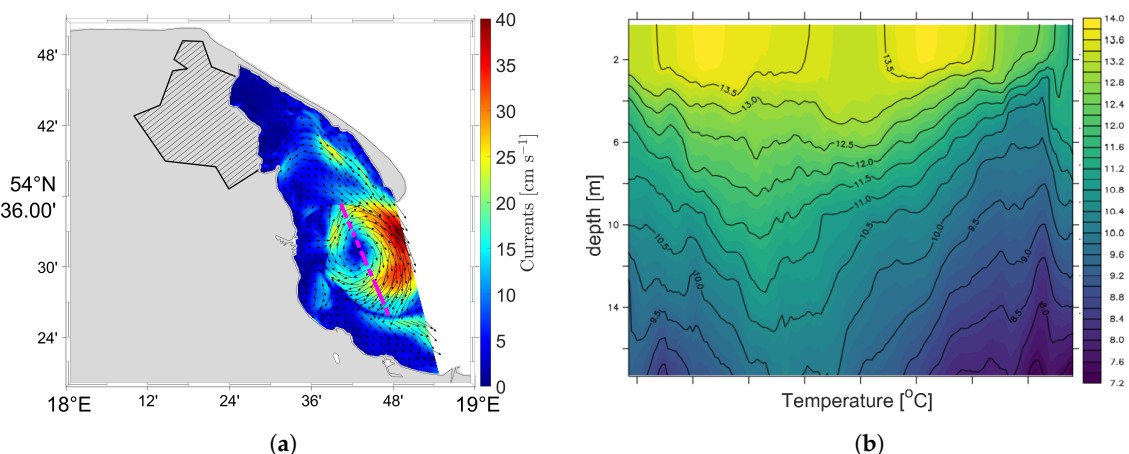

| (a) | (b) |

**Figure 15.** Surface currents distribution of anticyclone on 24 May 2018 (**a**) with marked (dashed purple) line of vertical cross-section of water temperature (**b**).

Analyzing the distribution of currents in individual months (see Figures A21 and A22) in relation to sea surface height in those months (see Figures A19 and A20), we can observe some correlations. If the vectors of currents are directed in one direction for most cells in the domain, then the flow of water takes place in the direction indicated by these vectors and there are two possible consequences. The first one when the water is pushed into the bay (i.e., October 2016) and the second when the water is drained from the bay (i.e., December 2016). In the first case, sea surface height on the western side of the domain is higher than on the eastern side. In the second case the sea surface height is relatively lower on the western side of the domain than on the eastern side.

### 3.3. Web Portal

Presently, quick access to expert knowledge is highly valued, especially in the context of decision-making by the authorities [21]. In order to meet these requirements, the internet service has been developed. This is one of the key services created within the framework of the project, on which the results of all models included in the WaterPUCK Integrated Information and Prediction Service will be made available. This website will operate dynamically in the operational mode allowing for visualization of forecast maps, time series and vertical sections.

In this paper, we present the launch of the website development version, which allows you to generate maps of selected hydrodynamic parameters from the EcoPuckBay model in the area of the Puck Bay and the western part of the Gulf of Gdańsk. Currently, service has a temporal coverage from 2014 to 2018. It is now possible to generate raster maps (Figure 16a) for individual depths, which represent the next vertical level of the model. In addition, it is possible to create time series (Figure 16b) for set periods in the selected location (after prior determination or indication of the desired latitude and longitude) as well as W-E and/or S-N cross-sections, allowing for analysis of the variability of the parameter state in the entire water column (Figure 16c). This service can be accessed from the project website (www.waterpuck.pl) after choosing "Services" from the navigation menu and selecting "EcoPuckBay Hydrodynamical Model of the Puck Bay".

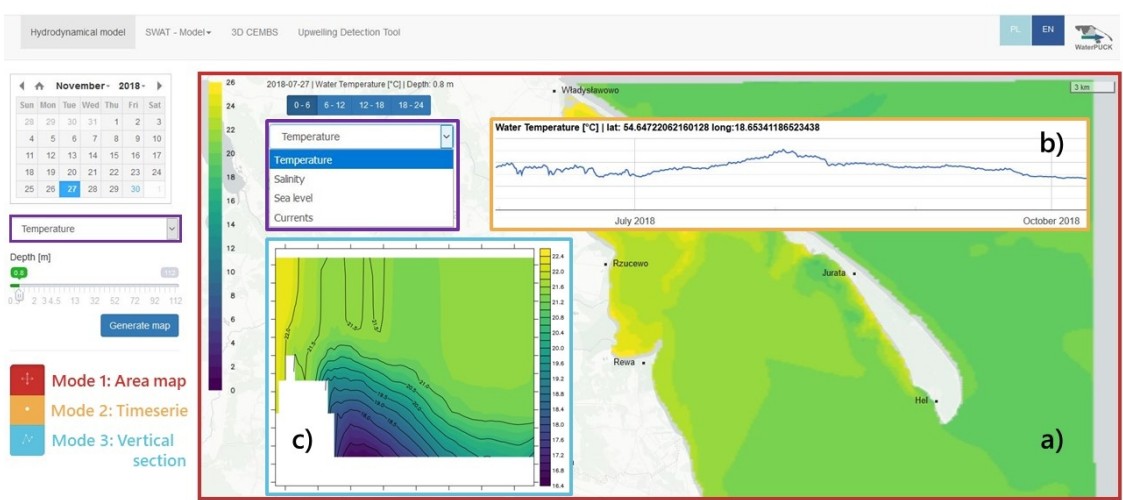

**Figure 16.** Screenshot of navigation menu and results for the Web Portal service (**a**) raster map (**b**) time series (**c**) vertical cross-section of selected parameter.

The portal operates on the basis of front-end/back-end technology. The mechanisms responsible for displaying (front-end) are separated from the data processing (back-end). This solution allows you to build a portal with high scaling capabilities, i.e., increasing the number of connections supported at one time. In addition, separation of displaying from data processing provides additional server security and allows independent development of both parts. Front-end layer responsible for data visualization was created in Bootstrap technology and adapted to be operated from mobile devices. Back-end responsible for data processing and transferring them to the front-end part was created in the RESTful-API technology. This technology is based on communication between parts of the portal using stateless queries (Figure 17).

The hydrodynamic data comes from the EcoPuckBay model run on the Tryton computing cluster in CI TASK [22] once a day. After completing the calculations, the data is copied to the disk space of the back-end part.

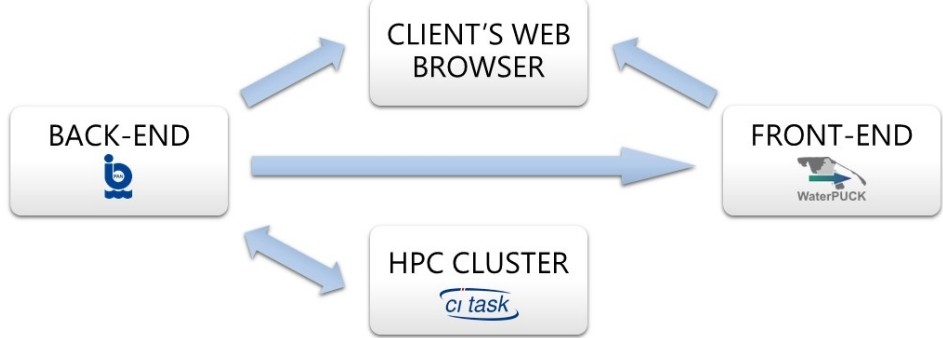

**Figure 17.** Architecture diagram of the WaterPUCK project's service - the hydrodynamic part of the EcoPuckBay model.

## 4. Discussion

The paper presents the hydrodynamic part of the three-dimensional ecohydrodynamic model of the Puck Bay EcoPuckBay, which determines the main physical environmental parameters such as water potential temperature, salinity, horizontal velocity components and sea surface height.

The statistical analysis was carried out to confirm that EcoPuckBay model's predictions are similar to field data. The time series from model simulations were compared with in situ samples from several locations across the entire effective domain using VIEP environmental samples database and s/y Oceania vertical profiles conducted during its cruise. For all locations described in the paper, we calculated basic statistics such as RMSE, standard deviations (STD) and Pearson's correlation coefficients for the full period of available measurements in the dataset (Section 3.1).

The comparison of the EcoPuckBay model computations revealed that modeled water temperature results are in very good agreement with experimental samples with lowest correlation of 0.92 for s/y Oceania profiles and 0.97 when tested against time series of VIEP samples (Table 3). To balance our model against another state-of-the-art model, we compared numerical data from NEMO-Nordic to VIEP time series measurements. The Pearson's correlation coefficient between NEMO-Nordic model and VIEP data is equal to 0.98. On the one hand, this comparison shows that the temperature is slightly better reproduced by the NEMO-Nordic model. On the other hand, taking into account the salinity (Table 6) we see a significant advantage of the EcoPuckBay model. This is most likely due to the higher resolution of the EcoPuckBay model compared to the NEMO-Nordic model resolution. It is worth mentioning that the Pearson's correlation coefficients for vertical profiles of salinity are in the range from 0.78 to 1.0, with a value equal to 0.9 for all stations. However, for VIEP time series data we have a correlation coefficient equal to 0.58 for all stations.

It is worth mentioning that the shape of the mixed layer depends on vertical grid resolution as well as on vertical viscosity. Vertical mixing scheme provides the best solutions if the horizontal resolution is not variable. It means if our model will have a constant thickness of the cells, vertical profiles will better fit observation data. Bathymetry of the model has been adapted to the Puck Bay and not for the Gdańsk Bay, therefore from the depth of 4 m the thickness of layers increases up to 5 m, which in turn affects the shape of temperature and salinity profiles. This choice is a consequence of the trade-off between the vertical resolution and computational time. Adopting the same cell thicknesses for the entire domain as for Puck Bay would lead to over 250 layers, which would consequently lengthen the calculations by 8 times. The POP model which is the main core of the hydrodynamic part has been purposed for global simulations. Adopting it for shallow waters required modifying the dependence of vertical viscosity on Richardson's number. Our modification (suggested by Durski et al. [7]) brought the adopted turbulence scheme closer to the parameterization of Mellor-Yamada (level 2.5) which is one of the best for estuaries and shelf seas.

Analyzing the 5-year period (2014–2018), the greatest variability of surface temperature can be observed inside Puck Lagoon. In summer the surface temperature inside Puck Lagoon is about

5 °C higher than in the rest of the Puck Bay and in winter about 3 °C lower (Figures A8 and A9). Such changes are due to the geomorphological separation of Puck Lagoon from the rest of the Puck Bay and to the fact that the area of Puck Lagoon is 30% of the entire Puck Bay and only about 6% of the water volume of the entire Puck Bay is located within Puck Lagoon.

Fresh water inflow from rivers (mainly from the Vistula river) has the greatest influence on the variability of surface salinity values in the Puck Bay. Analyzing the modeled period, we can see that in the summer the surface salinity decreases most strongly in the vicinity of the Vistula river and inside the Puck Lagoon (Figures A17 and A18). The changes in the south part of domain are a consequence of the natural cycle of the Vistula discharges. Within Puck Lagoon, isolation from the rest of the Puck Bay and the discharge of fresh water from rivers 8–13 (mainly from Reda, see Figure 1) has the most impact on the variability of the surface salinity.

The variability of the monthly sea surface height average in the studied period is in line with expectations and its gradient is similar to the wind forcing fields (Figures A19 and A20). Average monthly values of currents, depending on the month, varied from about 2 cm·s$^{-1}$ to over 4.5 cm·s$^{-1}$ and were higher for winter months in relation to summer months (Table A13).

The analysis show that the dynamics of changes is well reflected in general. Standard deviation was quite low and systematic error was negative in comparison with, which means that model results were usually lower than the results from VIEP database. The analyzed period 2014 to 2018 can be characterized as moderate dynamic in terms of inter-annual variability of parameters. The correspondence between EcoPuckBay model results and observations in terms of both temporal and spatial analysis is encouraging. It is clearly visible that even though the model tends to slightly underestimate water temperature conditions it reacts very well to atmosphere forcing.

In high-resolution Baltic Sea configurations, river runoff is prone to cause numerical problems because the salinity is close to zero in the vicinity of some rivers. This can, at times, due to the dispersive nature inherent to all state-of-the-art advection schemes (other than the outdated upstream scheme), induce negative salinities for which the equation of state is not defined. We solved this problem by distributing the runoff over the 22 grid boxes closest to the actual position of the river mouth. The distribution among the grid cells is determined by weights calculated with a trigonometric decay radius of  2.5 km as a function of the respective cell distance to the location of the Vistula river mouth.

## 5. Conclusions

The paper describes the initial implementation of the EcoPuckBay model which is the part of the WaterPUCK project. The WaterPUCK project's goal is to provide integrated information and predictive service for the Puck District by developing a system that works in an operational state and providing reliable forecasts of the waters surrounding the region. The evaluation of the quality of the results obtained from the EcoPuckBay model for the period of January 2014 to December 2018 was performed by statistical comparison with in situ measurements and another model reanalysis.

The results of the model verification indicate its suitability for forecasting hydrodynamic conditions within the concerned region. Satisfactory compatibility between in situ measurements and simulations enables reliable physical conditions to be established for future simulations with the active biogeochemical part.

The correct representation of the mixing in the whole water column, advection and heat exchange, which control the heating and cooling of water masses, is of particular importance for the forecasting of biological processes especially in a region such as the Puck Bay, with its unique ecosystem, different from the conditions of the open sea due to the presence of natural topographical barriers as well as the strong influence of environmental factors, both natural and induced by human activity.

The presented results show that the analyzed effective domain model dynamically reacts to the considered forces both in the coastal zone and further southeast towards the Gulf of Gdańsk, which depends on the changes in the open Baltic Sea.

The lower accuracy of the results that can be observed on some of the measuring stations may be due to the fact that some modules of the final product are still under development or testing, as well as the nature of the samples used for model evaluation.

Visible improvement of the EcoPuckBay model results is expected when the satellite and environmental data assimilation module is enabled.

The current configuration also uses forcings, which has been prepared from long-term averages (e.g., fresh water from water courses flowing into the Puck Bay) replacing target methods, which will be characterized by much higher time resolution of the information provided and will have a correctly mapped character and dynamics of individual external forcings. An example of a product that is currently being replaced by climatological averages is the SWAT hydrological model, which will provide information about the volume and temperature of fresh water and the concentrations of deposited inorganic substances entering the Puck Bay together with surface waters.

To study the complexity of hydro-physical and biological processes in the marine environment, and the links between these processes, modern techniques, i.e., mathematical modeling and computer simulations are required. Although the field work provides the most reliable information on these mechanisms and processes, it requires comprehensive and costly in situ observations conducted under a variety of hydrological conditions for long periods of time.

It is also worth mentioning that in situ measurements conducted at the monitoring stations have the highest accuracy of all monitoring method; however readings and the samples collected during the monitoring process are only snapshots of the environment state which in fact is the 1-dimensional 'here and now' information. In addition there is a limited quantity of data for each parameter at selected station with irregular time span between the campaigns.

The development of integrated approaches, such as monitoring measures and modeling, became an important tool not only for understanding the processes taking place in both inland and marine ecosystems but also for evaluating the impact of various land-use and climate scenarios on water quantity and quality at the basin scale. The main objective of this paper, i.e., development of a hydrodynamic (physical) module for the EcoPuckBay model, was achieved in the following steps:

- down-scaling and adapting the 3D CEMBS model for the Puck Bay region;
- advanced model tuning and optimization;
- topography implementation on the model grid;
- production of atmospheric data based on the UM weather model;
- incorporation of rivers from hydrological model SWAT;
- opening the boundary with the Baltic Sea;
- switching to the operational mode

Along with the operational mode that will be introduced in the EcoPuckBay model within upcoming development stages, an assimilation module will be enabled in the final configuration. This module's job is to process all the available environmental and satellite data at the time of the forecast start and force the simulation to produce results that better fit the environment state. This will provide lower model uncertainties and therefore improve overall accuracy.

Satellite–measured sea surface temperature (SST) information for the assimilation module will be taken from the Moderate Resolution Imaging Spectroradiometer (MODIS, Aqua satellite). Environmental packages of in situ data samples collected during VIEP monitoring campaigns will be delivered once per year. On a delivery event EcoPuckBay model will induce the 1-year hindcast parallel simulation that will assimilate those results and after reaching present day updating the archives and switching back to the operational state continuing with forecasts.

**Author Contributions:** Conceptualization, D.D. and L.D.-G.; Methodology, D.D., J.J., M.J. and L.D.-G.; Software, D.D. and A.N.; Validation, D.D.; Formal analysis, D.D. and L.D.-G.; Investigation, D.D. and J.J.; Resources, D.D. and J.J.; Data curation, D.D., J.J. and D.R.; Writing—original draft preparation, D.D., M.J. and L.D.-G.; Writing—review and editing, D.D., M.J. and L.D.-G.; Visualization, D.D.; Supervision, L.D.-G.; Project administration, L.D.-G.; Funding acquisition, L.D.-G.

**Funding:** This research was funded by National Centre for Research and Development of Poland within the BIOSTRATEG III program No. BIOSTRATEG3/343927/3/NCBR/2017.

**Acknowledgments:** This study has been conducted using E.U. Copernicus Marine Service Information. Calculations were carried out at the Academic Computer Centre in Gdańsk.

**Conflicts of Interest:** The authors declare no conflict of interest.

## Appendix A

Main part of the paper was limited to the monthly averages and statistical comparison results. However, detailed Figures and Tables for separate years and measurement stations are presented in the Appendix section.

*Appendix A.1. Temperature*

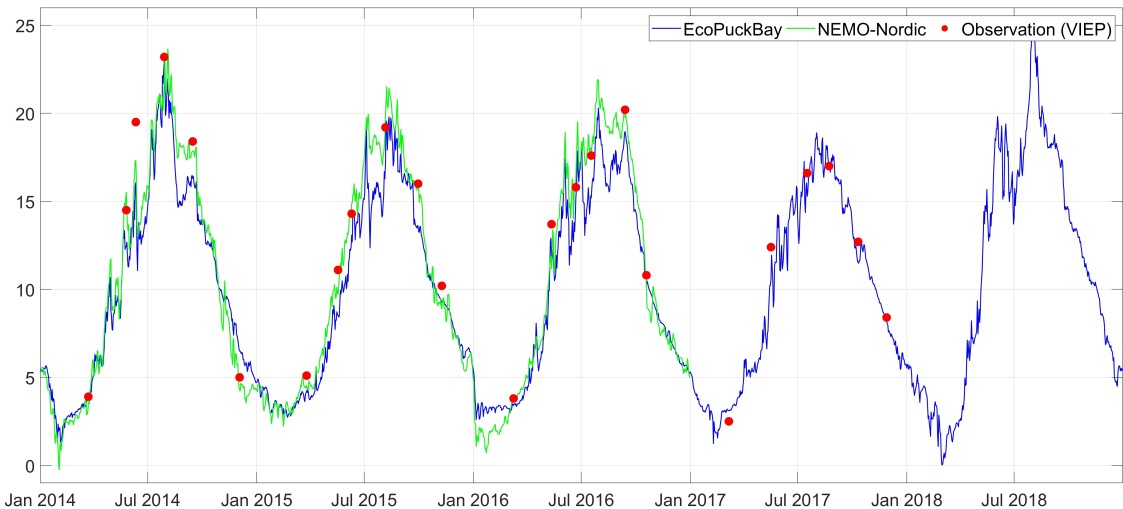

**Figure A1.** Time series of surface water temperature on station OM1 compared with VIEP observations and NEMO-Nordic model.

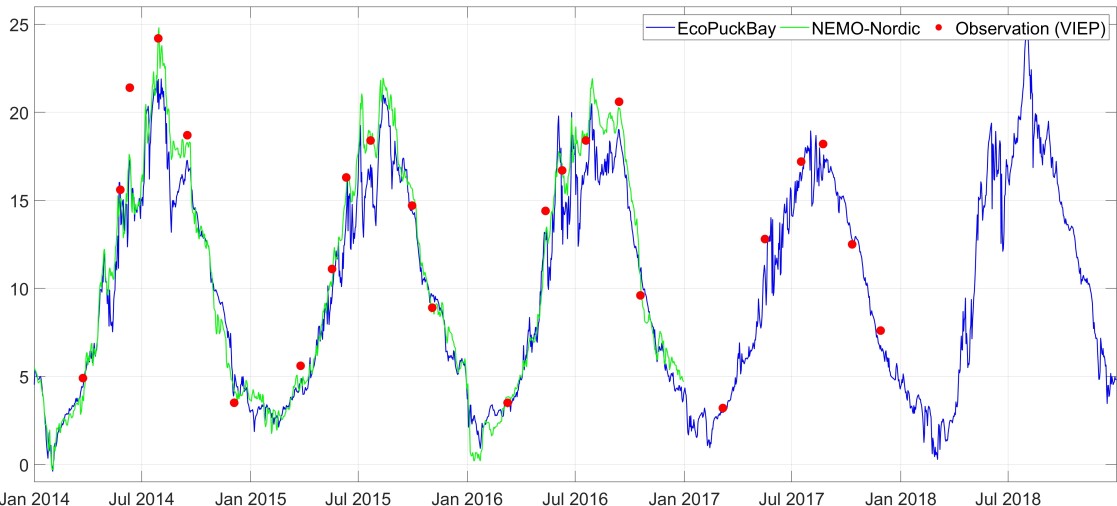

**Figure A2.** Time series of surface water temperature on station T11 compared with VIEP observations and NEMO-Nordic model.

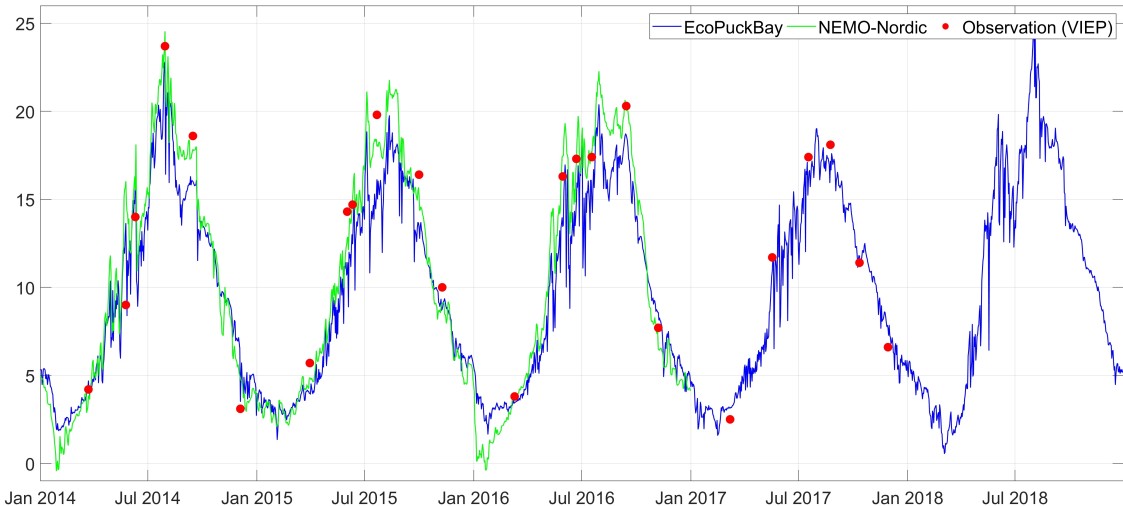

**Figure A3.** Time series of surface water temperature on station T12 compared with VIEP observations and NEMO-Nordic model.

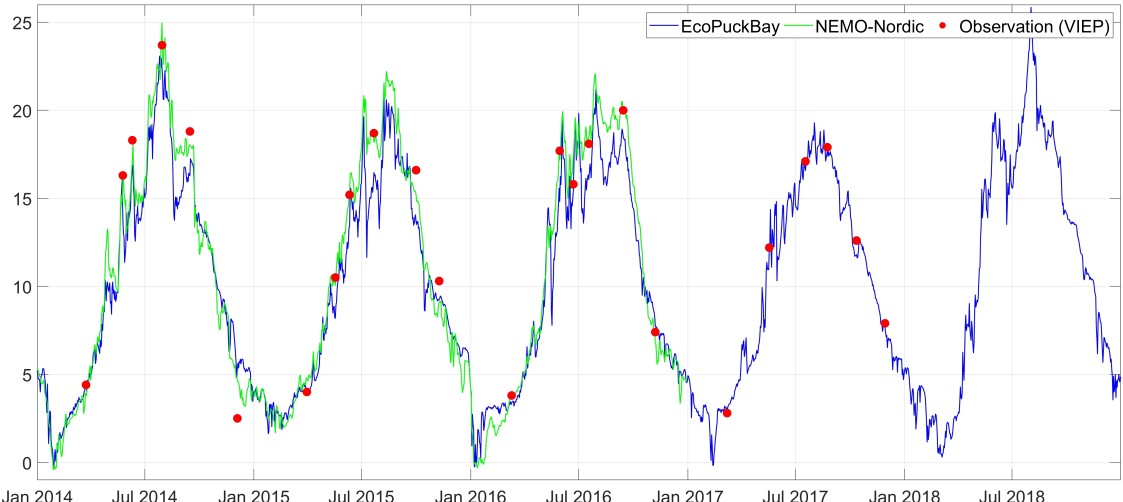

**Figure A4.** Time series of surface water temperature on station T14 compared with VIEP observations and NEMO-Nordic model.

**Table A1.** Monthly means for surface temperature for the years 2014–2018.

| Year\Month | Jan | Feb | Mar | Apr | May | Jun | Jul | Aug | Sep | Oct | Nov | Dec |
|---|---|---|---|---|---|---|---|---|---|---|---|---|
| 2014 | 3.50 | 2.03 | 3.96 | 7.83 | 11.34 | 14.81 | 19.82 | 18.44 | 16.15 | 12.61 | 8.77 | 4.89 |
| 2015 | 3.38 | 2.70 | 3.97 | 5.78 | 10.13 | 14.19 | 16.59 | 18.82 | 16.41 | 11.67 | 8.89 | 6.37 |
| 2016 | 2.69 | 3.16 | 3.79 | 6.84 | 12.25 | 16.11 | 17.44 | 17.89 | 17.63 | 11.72 | 7.49 | 5.33 |
| 2017 | 3.02 | 2.32 | 3.83 | 6.30 | 10.87 | 14.32 | 16.95 | 18.40 | 16.12 | 12.68 | 9.04 | 5.80 |
| 2018 | 4.03 | 2.34 | 1.67 | 6.44 | 14.23 | 17.97 | 19.73 | 21.25 | 17.81 | 13.18 | 9.67 | 5.38 |
| mean | 3.32 | 2.51 | 3.44 | 6.64 | 11.76 | 15.48 | 18.11 | 18.96 | 16.82 | 12.37 | 8.77 | 5.55 |

**Table A2.** Monthly minimums for surface temperature for the years 2014–2018.

| year\month | Jan | Feb | Mar | Apr | May | Jun | Jul | Aug | Sep | Oct | Nov | Dec |
|---|---|---|---|---|---|---|---|---|---|---|---|---|
| 2014 | 0.59 | −0.16 | 3.54 | 6.45 | 9.55 | 12.83 | 17.78 | 17.30 | 15.45 | 11.19 | 5.76 | 0.74 |
| 2015 | 0.42 | 0.91 | 3.69 | 4.74 | 8.61 | 12.72 | 15.19 | 17.43 | 15.30 | 9.38 | 6.00 | 3.51 |
| 2016 | −0.28 | 2.23 | 3.59 | 5.69 | 10.33 | 13.62 | 16.01 | 16.93 | 17.12 | 9.11 | 3.70 | 2.29 |
| 2017 | 0.15 | 0.47 | 3.45 | 4.98 | 8.96 | 12.62 | 15.17 | 16.88 | 15.46 | 11.58 | 5.27 | 1.64 |
| 2018 | 0.34 | −0.04 | 0.32 | 5.20 | 11.05 | 15.50 | 18.02 | 19.87 | 16.99 | 10.97 | 5.91 | 0.74 |

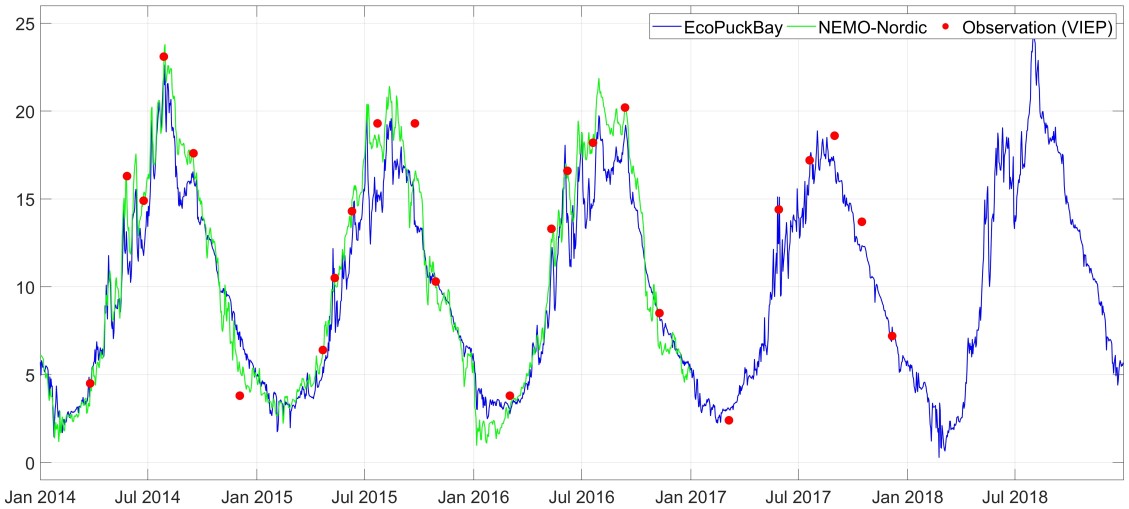

**Figure A5.** Time series of surface water temperature on station T16 compared with VIEP observations and NEMO-Nordic model.

**Figure A6.** Time series of surface water temperature on station ZG compared with VIEP observations and NEMO-Nordic model.

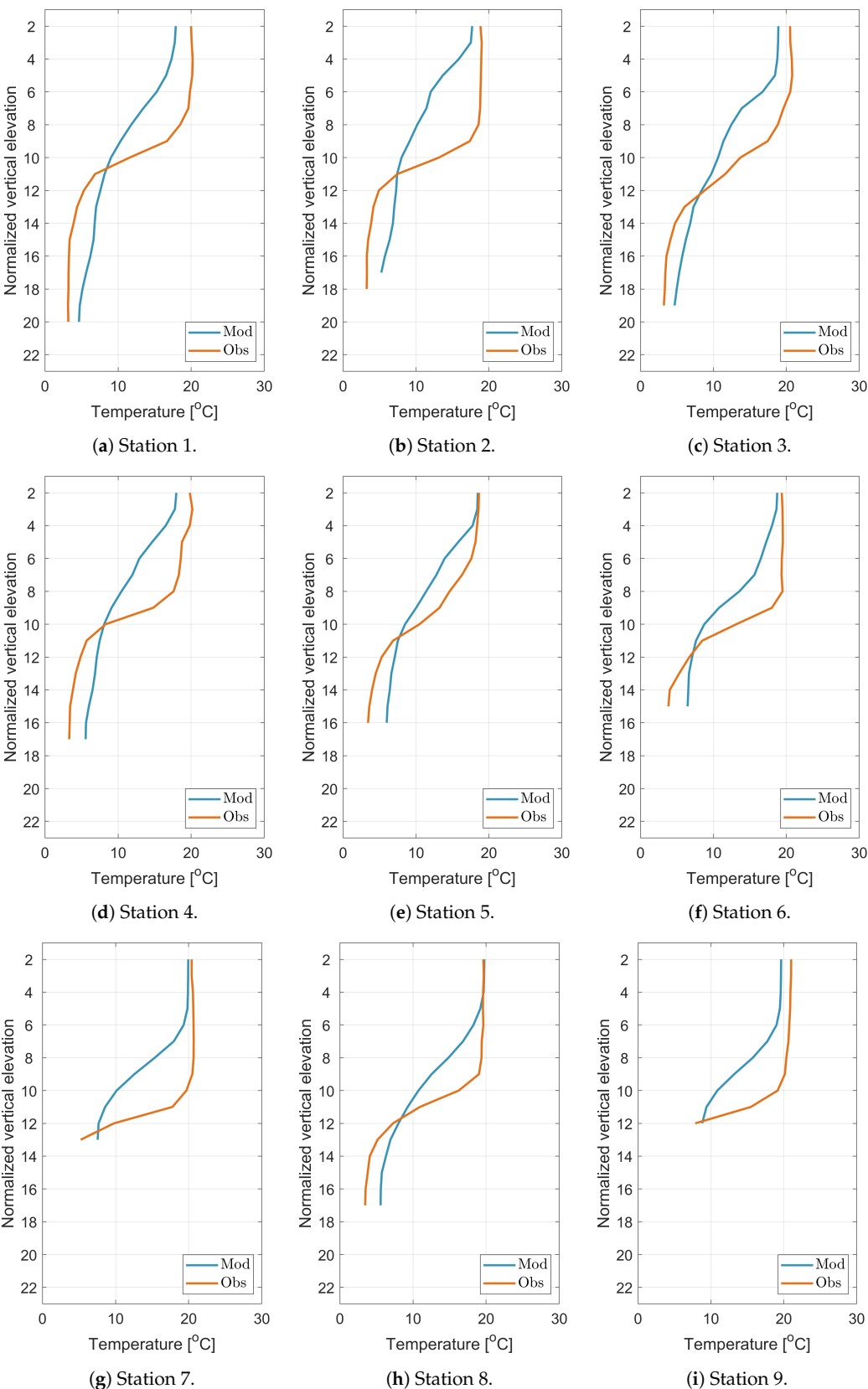

(**a**) Station 1.  (**b**) Station 2.  (**c**) Station 3.

(**d**) Station 4.  (**e**) Station 5.  (**f**) Station 6.

(**g**) Station 7.  (**h**) Station 8.  (**i**) Station 9.

**Figure A7.** *Cont.*

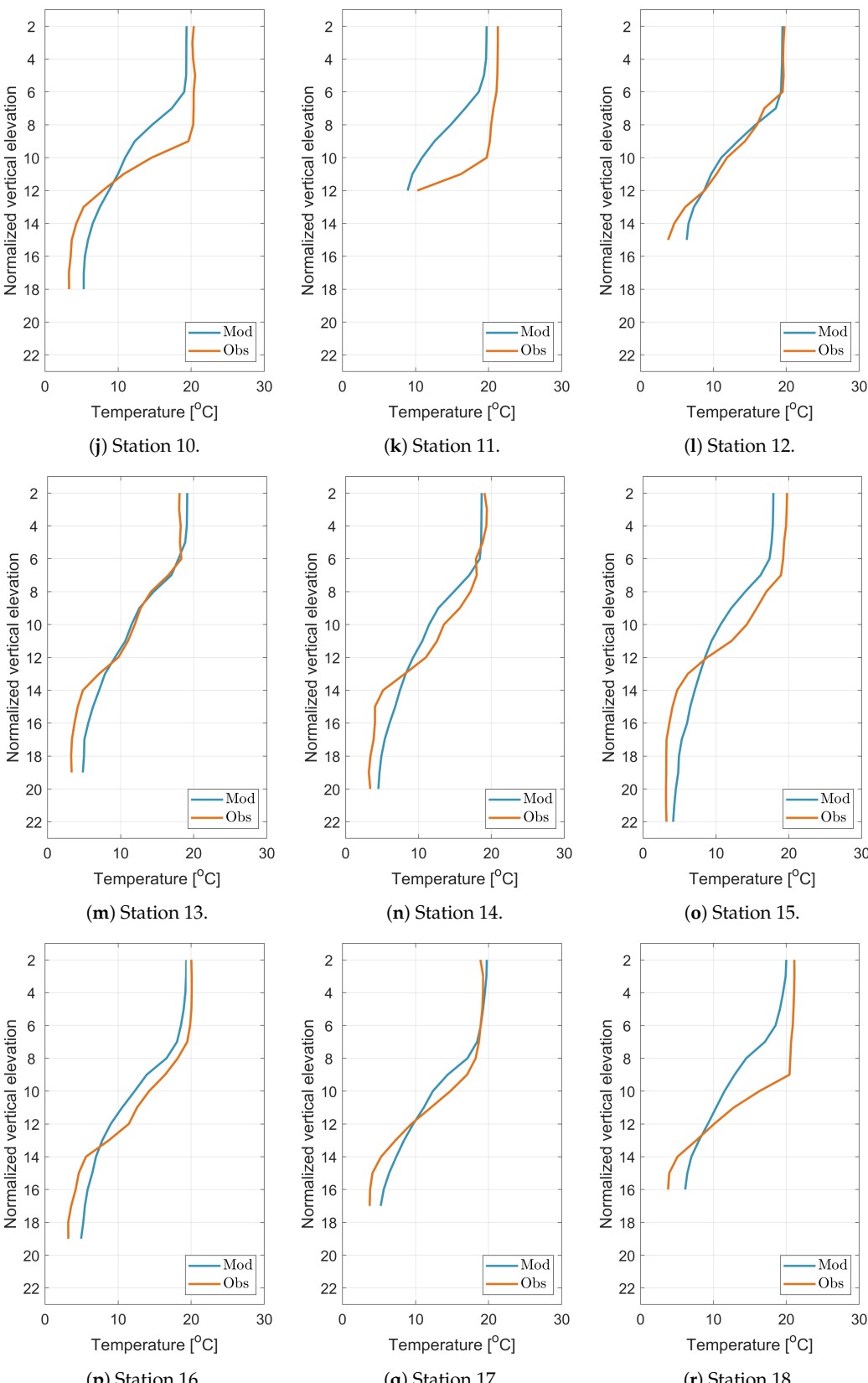

**Figure A7.** Temperature vertical profiles for all stations compared with s/y Oceania observations.

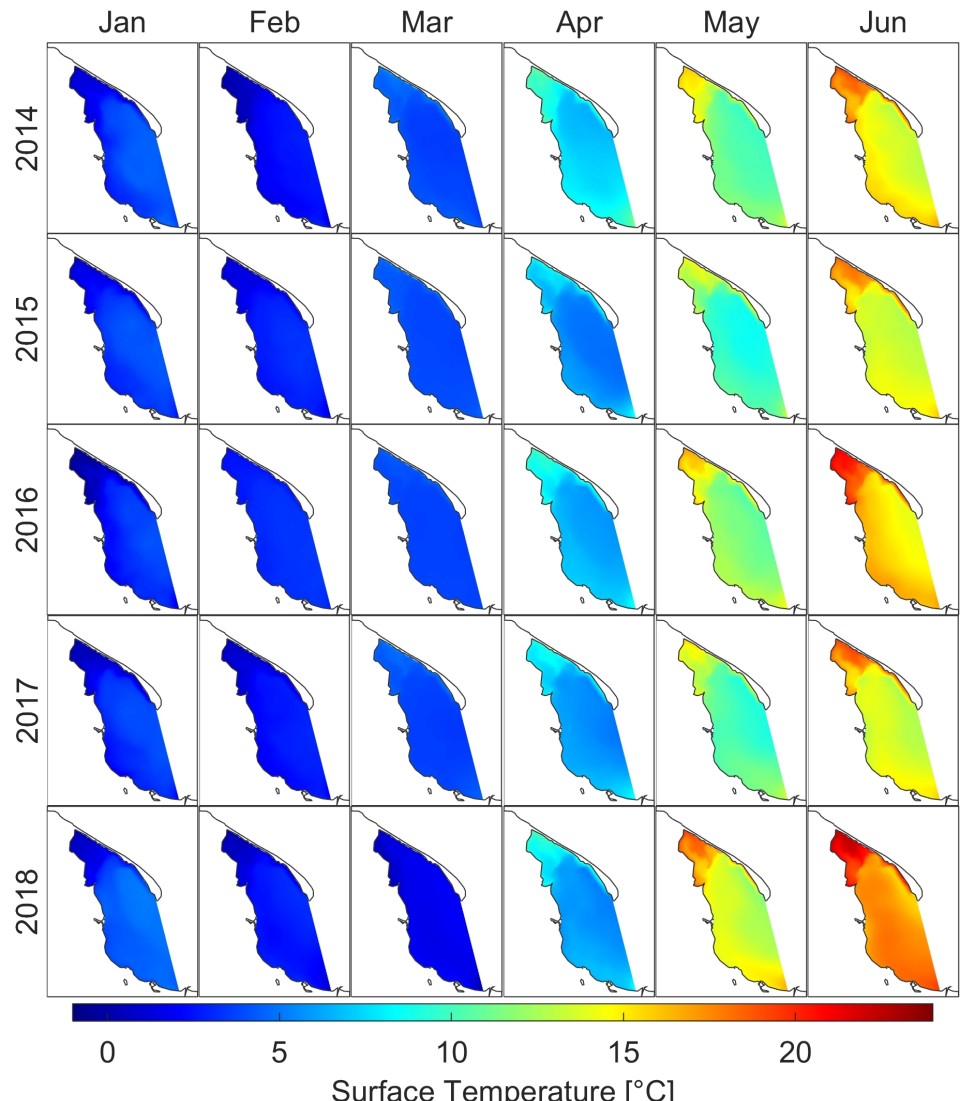

**Figure A8.** Monthly means of water temperature at sea surface for separated years. January to June.

**Table A3.** Monthly maximums for surface temperature for the years 2014–2018.

| year\month | Jan | Feb | Mar | Apr | May | Jun | Jul | Aug | Sep | Oct | Nov | Dec |
|---|---|---|---|---|---|---|---|---|---|---|---|---|
| 2014 | 4.65 | 2.85 | 5.60 | 11.63 | 16.13 | 19.57 | 24.21 | 21.14 | 17.74 | 13.26 | 9.56 | 6.37 |
| 2015 | 4.31 | 3.39 | 5.22 | 9.23 | 14.76 | 18.37 | 20.45 | 22.39 | 18.24 | 12.16 | 9.57 | 7.19 |
| 2016 | 4.15 | 3.50 | 4.67 | 10.18 | 17.34 | 21.50 | 21.88 | 20.24 | 18.93 | 12.29 | 8.53 | 6.44 |
| 2017 | 4.13 | 3.07 | 5.32 | 9.47 | 15.83 | 19.55 | 20.75 | 21.64 | 17.16 | 13.31 | 9.84 | 6.99 |
| 2018 | 5.15 | 3.25 | 2.56 | 10.84 | 19.56 | 22.35 | 24.03 | 23.89 | 18.98 | 13.74 | 10.58 | 6.55 |

**Table A4.** Monthly standard deviations for surface temperature for the years 2014–2018.

| year\month | Jan | Feb | Mar | Apr | May | Jun | Jul | Aug | Sep | Oct | Nov | Dec |
|---|---|---|---|---|---|---|---|---|---|---|---|---|
| 2014 | 1.01 | 0.71 | 0.38 | 1.00 | 1.42 | 1.41 | 1.20 | 0.86 | 0.43 | 0.30 | 0.79 | 1.34 |
| 2015 | 0.83 | 0.58 | 0.22 | 0.95 | 1.43 | 1.27 | 1.11 | 1.02 | 0.55 | 0.52 | 0.71 | 0.76 |
| 2016 | 1.21 | 0.27 | 0.17 | 0.94 | 1.36 | 1.67 | 1.28 | 0.74 | 0.37 | 0.61 | 1.07 | 0.91 |
| 2017 | 1.03 | 0.60 | 0.35 | 0.91 | 1.37 | 1.47 | 1.16 | 1.06 | 0.39 | 0.34 | 1.02 | 1.24 |
| 2018 | 1.20 | 0.69 | 0.30 | 1.03 | 1.55 | 1.40 | 1.24 | 0.86 | 0.41 | 0.51 | 0.97 | 1.39 |

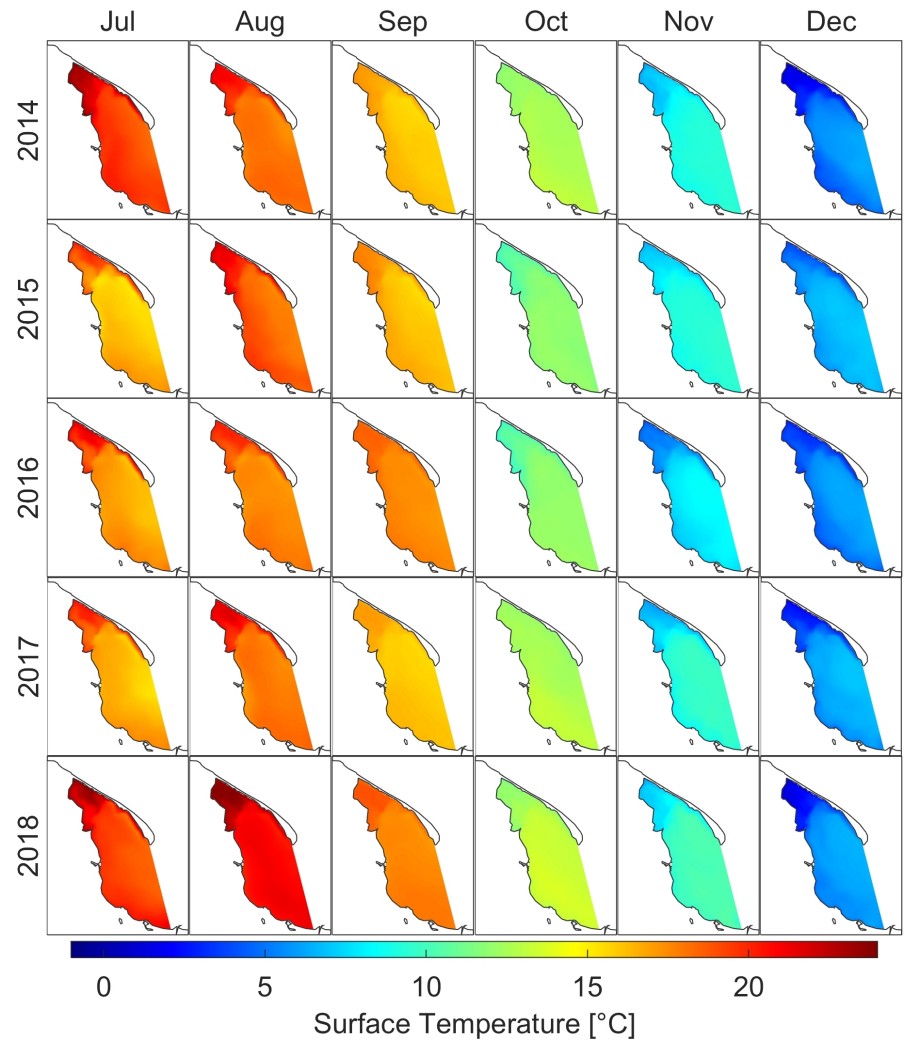

**Figure A9.** Monthly means of water temperature at sea surface for separated years. July to December.

*Appendix A.2. Salinity*

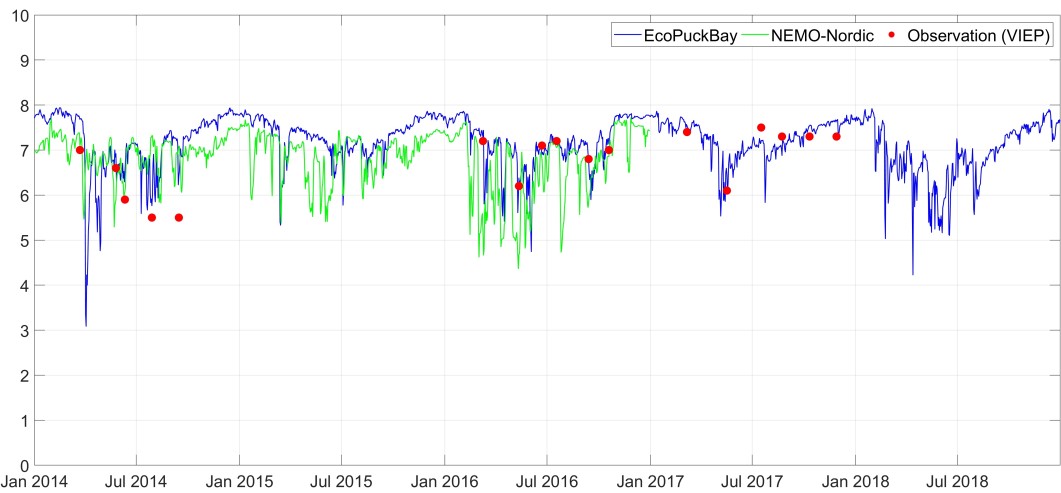

**Figure A10.** Time series of surface salinity on station OM1 compared with VIEP observations and NEMO-Nordic model.

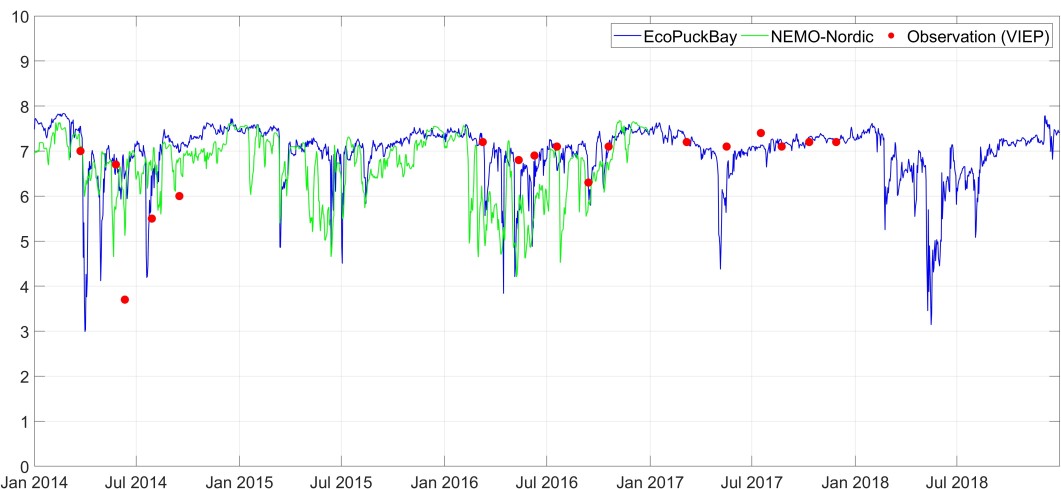

**Figure A11.** Time series of surface salinity on station T11 compared with VIEP observations and NEMO-Nordic model.

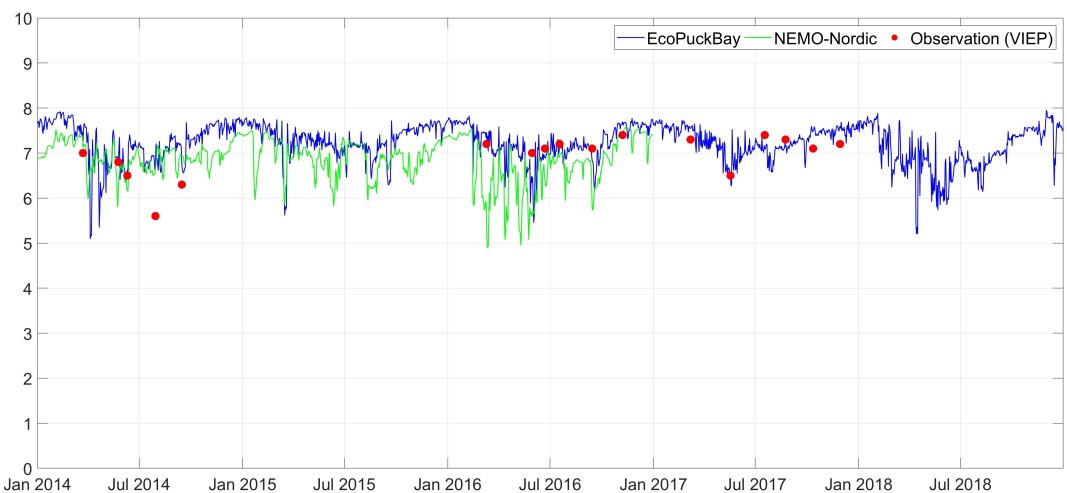

**Figure A12.** Time series of surface salinity on station T12 compared with VIEP observations and NEMO-Nordic model.

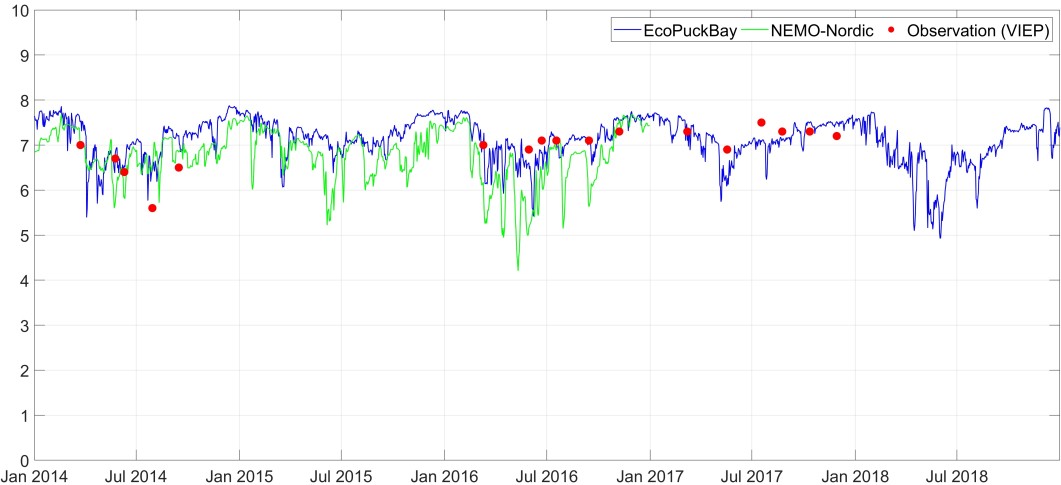

**Figure A13.** Time series of surface salinity on station T14 compared with VIEP observations and NEMO-Nordic model.

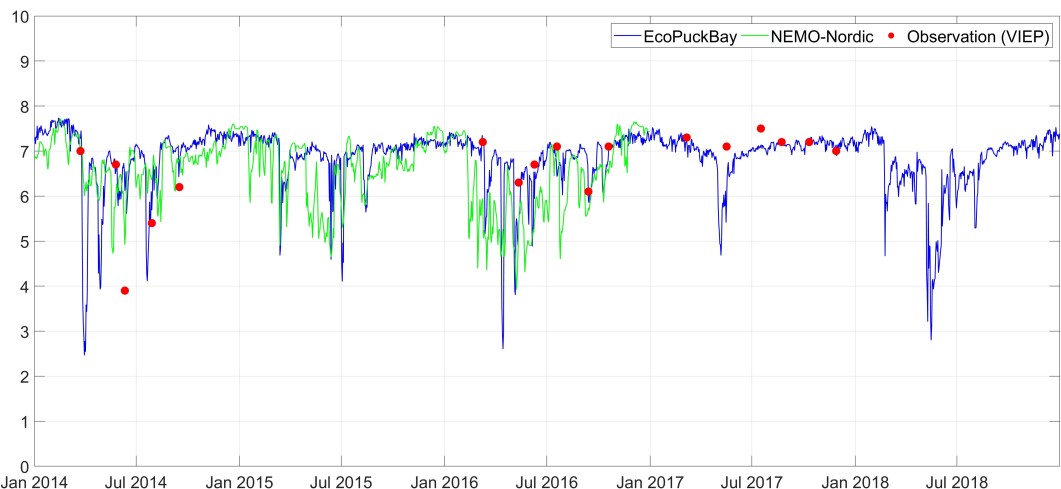

**Figure A14.** Time series of surface salinity on station T16 compared with VIEP observations and NEMO-Nordic model.

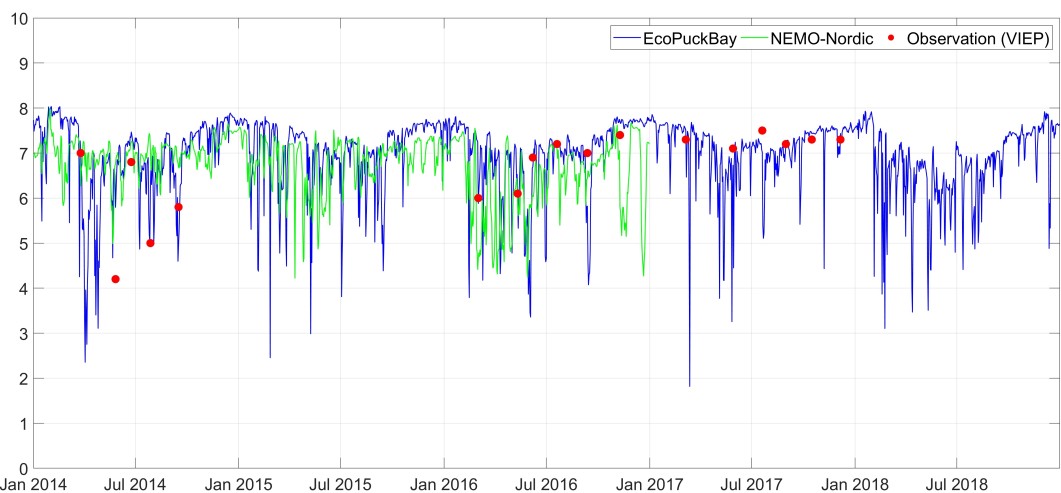

**Figure A15.** Time series of surface salinity on station ZG compared with VIEP observations and NEMO-Nordic model.

**Table A5.** Monthly means for surface salinity for the years 2014–2018.

| Year\Month | Jan | Feb | Mar | Apr | May | Jun | Jul | Aug | Sep | Oct | Nov | Dec |
|---|---|---|---|---|---|---|---|---|---|---|---|---|
| 2014 | 7.46 | 7.89 | 7.15 | 5.79 | 6.45 | 6.77 | 6.55 | 6.87 | 7.01 | 7.47 | 7.66 | 7.76 |
| 2015 | 7.62 | 7.43 | 6.79 | 7.23 | 7.00 | 6.69 | 6.91 | 6.87 | 7.07 | 7.47 | 7.68 | 7.67 |
| 2016 | 7.67 | 7.42 | 6.75 | 6.50 | 6.19 | 6.57 | 6.94 | 7.11 | 6.76 | 7.33 | 7.70 | 7.70 |
| 2017 | 7.68 | 7.59 | 7.03 | 6.86 | 6.06 | 6.97 | 7.09 | 7.08 | 7.17 | 7.51 | 7.55 | 7.51 |
| 2018 | 7.59 | 6.76 | 6.61 | 6.38 | 5.56 | 5.97 | 6.45 | 6.55 | 6.98 | 7.45 | 7.49 | 7.50 |
| mean | 7.60 | 7.42 | 6.87 | 6.55 | 6.25 | 6.59 | 6.79 | 6.90 | 7.00 | 7.45 | 7.62 | 7.63 |

**Table A6.** Monthly minimums for surface salinity for the years 2014–2018.

| Year\Month | Jan | Feb | Mar | Apr | May | Jun | Jul | Aug | Sep | Oct | Nov | Dec |
|---|---|---|---|---|---|---|---|---|---|---|---|---|
| 2014 | 0.31 | 0.27 | 0.18 | 0.38 | 1.02 | 1.08 | 1.66 | 1.95 | 2.05 | 2.26 | 1.23 | 0.46 |
| 2015 | 0.23 | 0.22 | 0.20 | 0.50 | 1.14 | 1.19 | 1.40 | 2.16 | 2.04 | 2.06 | 0.76 | 0.41 |
| 2016 | 0.25 | 0.19 | 0.19 | 0.57 | 0.68 | 1.38 | 1.39 | 1.84 | 1.79 | 1.93 | 0.83 | 0.31 |
| 2017 | 0.20 | 0.22 | 0.19 | 0.51 | 0.41 | 1.28 | 1.65 | 2.08 | 2.00 | 1.24 | 0.70 | 0.36 |
| 2018 | 0.28 | 0.22 | 0.22 | 0.46 | 0.22 | 0.69 | 0.63 | 1.82 | 1.55 | 1.48 | 0.99 | 0.46 |

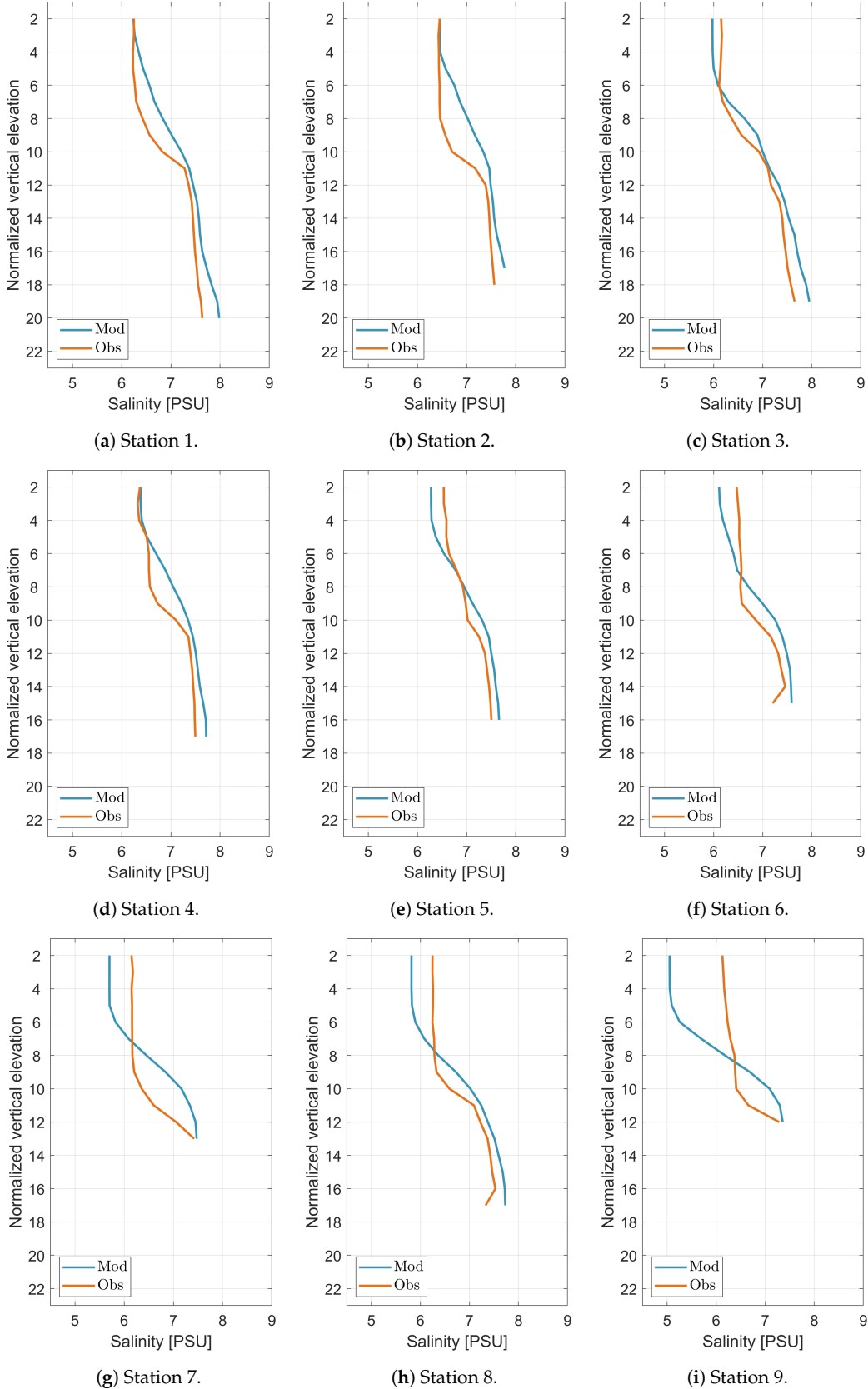

(**a**) Station 1.　　　(**b**) Station 2.　　　(**c**) Station 3.

(**d**) Station 4.　　　(**e**) Station 5.　　　(**f**) Station 6.

(**g**) Station 7.　　　(**h**) Station 8.　　　(**i**) Station 9.

**Figure A16.** *Cont.*

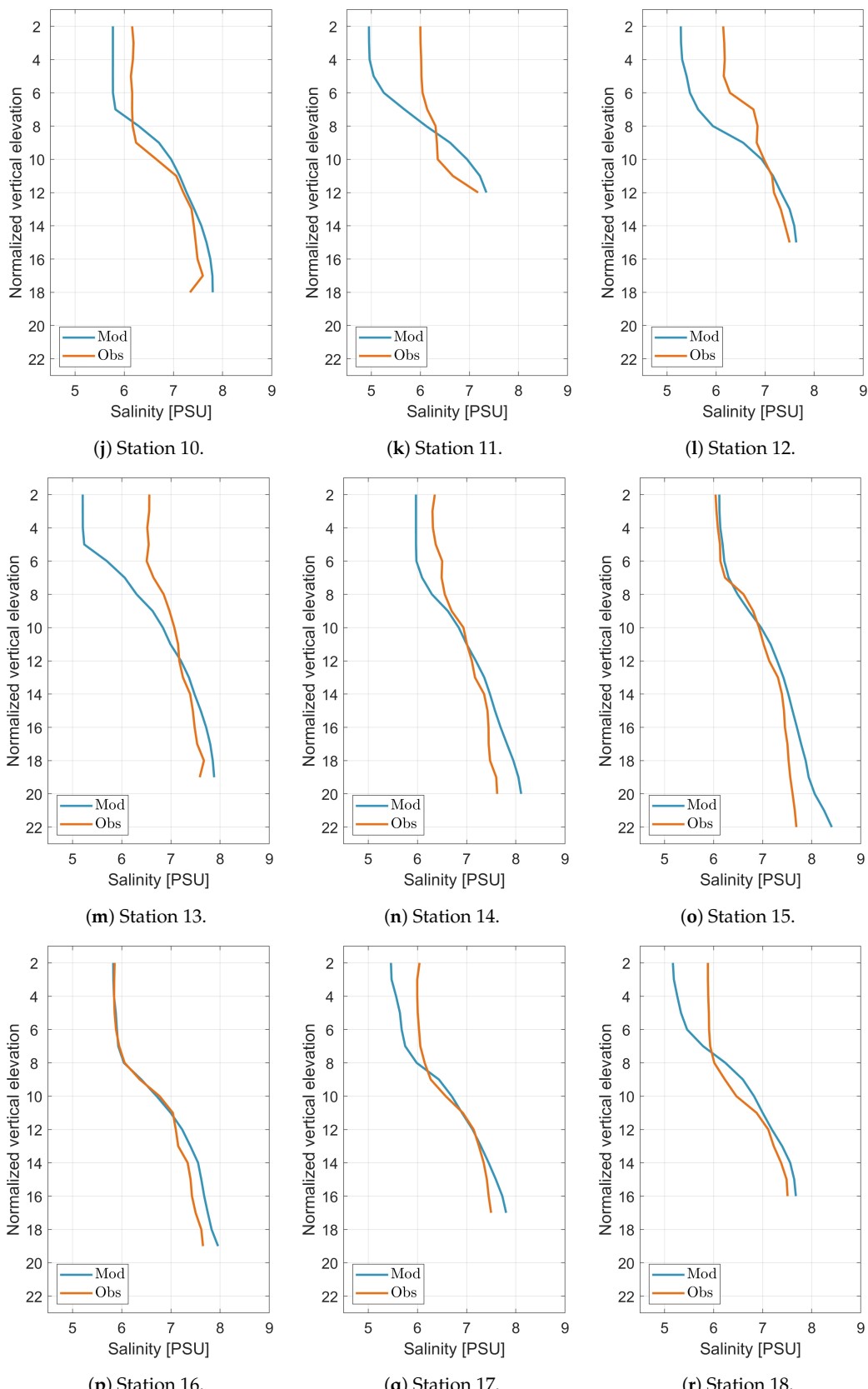

Figure A16. Salinity vertical profiles for all stations compared with s/y Oceania observations.

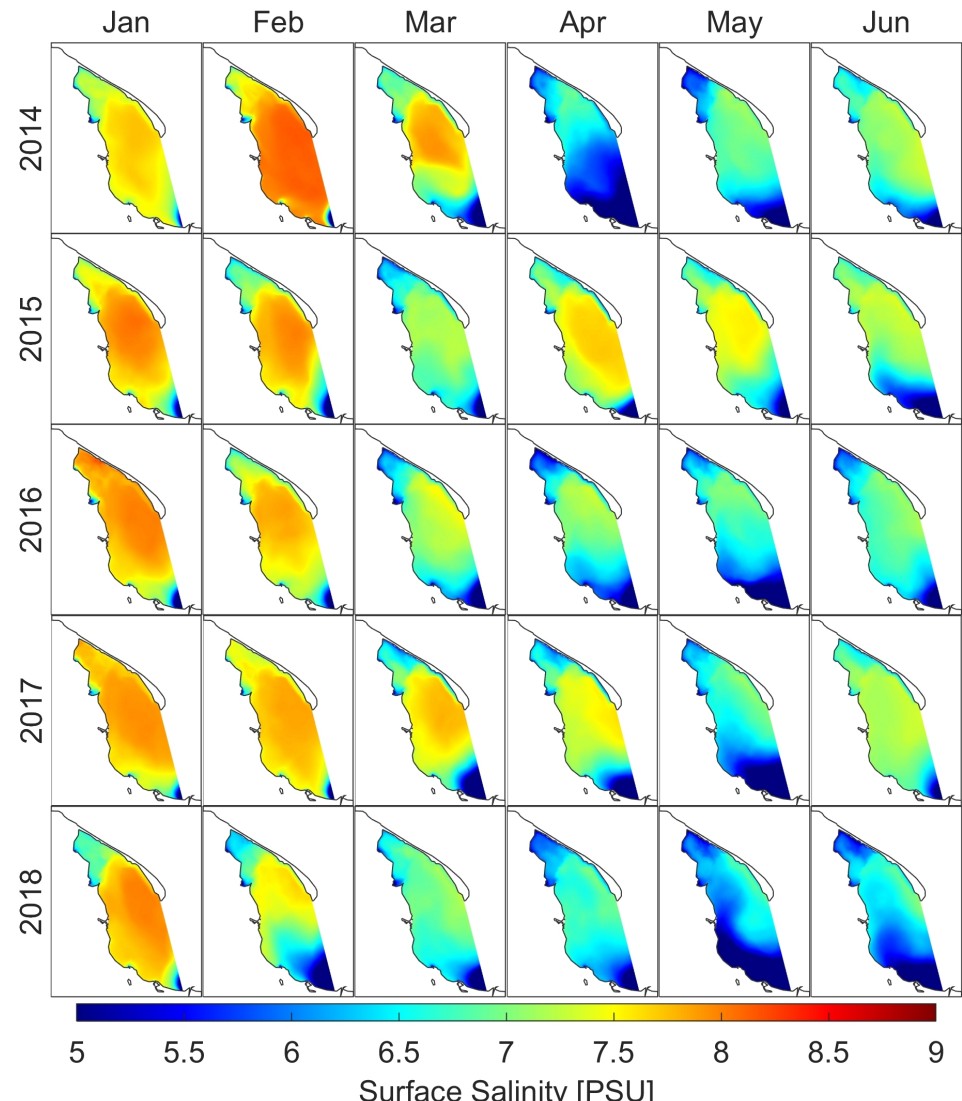

**Figure A17.** Monthly means of salinity at sea surface for separated years. January to June.

**Table A7.** Monthly maximums for surface salinity for the years 2014–2018.

| Year\Month | Jan | Feb | Mar | Apr | May | Jun | Jul | Aug | Sep | Oct | Nov | Dec |
|---|---|---|---|---|---|---|---|---|---|---|---|---|
| 2014 | 7.75 | 8.16 | 7.93 | 6.95 | 7.16 | 7.38 | 7.24 | 7.41 | 7.45 | 7.72 | 7.96 | 8.08 |
| 2015 | 8.07 | 8.00 | 7.27 | 7.71 | 7.56 | 7.38 | 7.41 | 7.31 | 7.44 | 7.74 | 7.97 | 8.06 |
| 2016 | 8.19 | 7.88 | 7.53 | 7.31 | 7.08 | 7.25 | 7.49 | 7.47 | 7.30 | 7.67 | 8.05 | 8.02 |
| 2017 | 7.98 | 7.86 | 7.81 | 7.63 | 7.16 | 7.36 | 7.35 | 7.45 | 7.55 | 7.75 | 7.89 | 7.94 |
| 2018 | 8.00 | 7.68 | 7.28 | 7.01 | 7.03 | 7.09 | 7.13 | 7.01 | 7.39 | 7.70 | 7.81 | 7.92 |

**Table A8.** Monthly standard deviations for surface salinity for the years 2014–2018.

| Year\Month | Jan | Feb | Mar | Apr | May | Jun | Jul | Aug | Sep | Oct | Nov | Dec |
|---|---|---|---|---|---|---|---|---|---|---|---|---|
| 2014 | 0.33 | 0.47 | 0.82 | 0.90 | 0.72 | 0.58 | 0.70 | 0.46 | 0.43 | 0.28 | 0.36 | 0.34 |
| 2015 | 0.48 | 0.68 | 0.57 | 0.67 | 0.65 | 0.83 | 0.52 | 0.55 | 0.40 | 0.30 | 0.27 | 0.36 |
| 2016 | 0.59 | 0.53 | 0.78 | 0.77 | 0.98 | 0.62 | 0.64 | 0.39 | 0.53 | 0.34 | 0.35 | 0.41 |
| 2017 | 0.40 | 0.43 | 1.06 | 0.89 | 1.05 | 0.45 | 0.39 | 0.47 | 0.51 | 0.26 | 0.34 | 0.41 |
| 2018 | 0.48 | 0.98 | 0.77 | 0.65 | 1.25 | 0.88 | 0.81 | 0.50 | 0.57 | 0.26 | 0.39 | 0.49 |

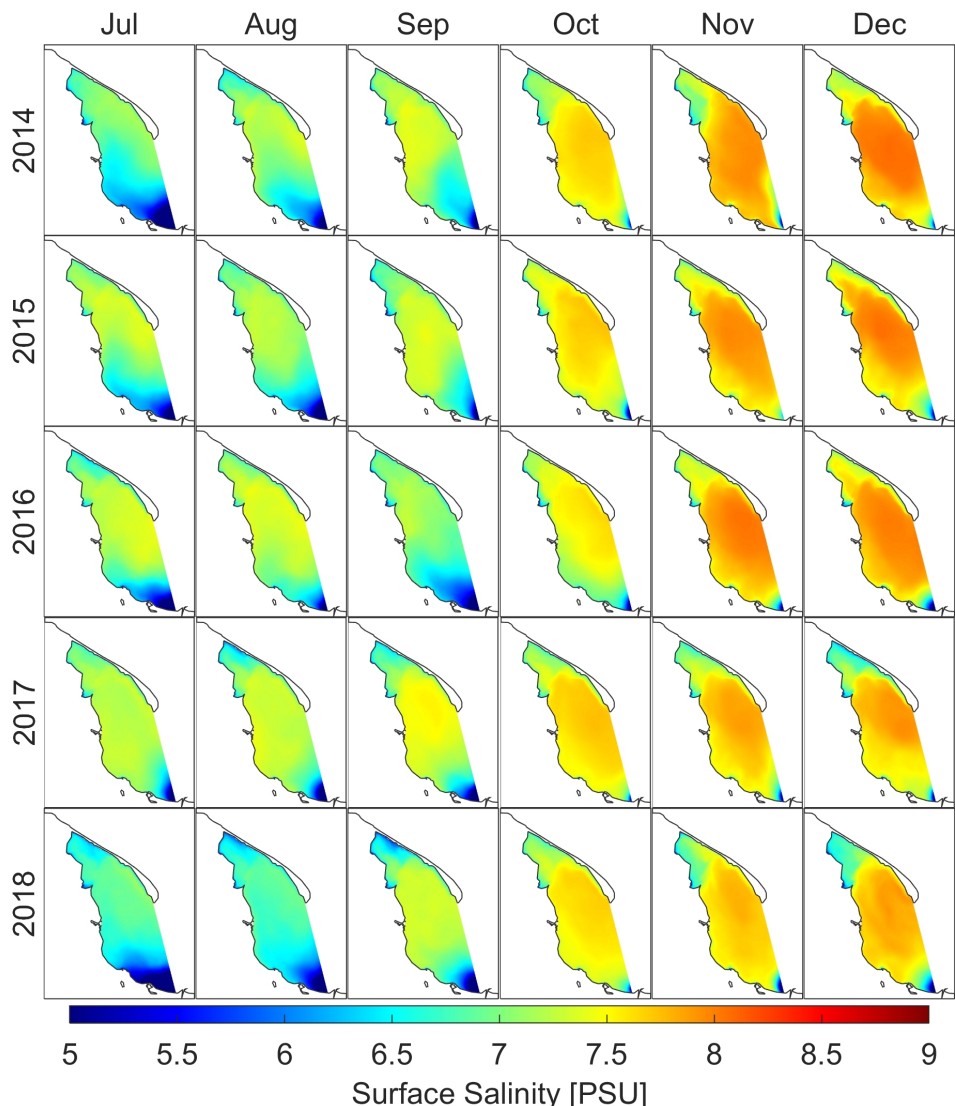

**Figure A18.** Monthly means of salinity at sea surface for separated years. July to December.

*Appendix A.3. Sea Surface Height*

**Table A9.** Monthly means for sea surface height for the years 2014–2018.

| Year\Month | Jan | Feb | Mar | Apr | May | Jun | Jul | Aug | Sep | Oct | Nov | Dec |
|---|---|---|---|---|---|---|---|---|---|---|---|---|
| 2014 | 0.28 | −0.37 | −0.52 | 0.29 | 0.49 | 0.26 | 0.72 | −0.16 | 0.02 | −0.01 | 0.07 | −1.03 |
| 2015 | −0.96 | −0.37 | 0.15 | −0.65 | −0.24 | −0.08 | −0.64 | 0.25 | −0.15 | 0.06 | −1.23 | −0.98 |
| 2016 | −0.43 | −0.54 | 0.04 | −0.02 | 0.41 | 0.33 | −0.04 | −0.43 | −0.23 | 0.90 | −0.26 | −1.14 |
| 2017 | −0.66 | −0.08 | −0.18 | −0.19 | 0.48 | −0.23 | −0.14 | −0.57 | −0.29 | −1.14 | −0.90 | −0.88 |
| 2018 | −0.30 | 0.16 | 0.87 | 0.49 | 0.93 | 0.32 | 0.44 | −0.22 | −0.87 | −0.64 | −0.15 | −0.75 |
| mean | −0.41 | −0.24 | 0.07 | −0.02 | 0.41 | 0.12 | 0.07 | −0.23 | −0.30 | −0.17 | −0.49 | −0.96 |

**Table A10.** Monthly minimums for sea surface height for the years 2014–2018.

| Year\Month | Jan | Feb | Mar | Apr | May | Jun | Jul | Aug | Sep | Oct | Nov | Dec |
|---|---|---|---|---|---|---|---|---|---|---|---|---|
| 2014 | −0.67 | −1.04 | −1.01 | −0.01 | −0.01 | −0.87 | 0.08 | −0.50 | −0.25 | −0.42 | −0.64 | −1.39 |
| 2015 | −1.57 | −0.60 | −0.14 | −2.66 | −0.74 | −0.68 | −1.27 | −0.22 | −0.42 | −0.37 | −2.00 | −1.40 |
| 2016 | −0.67 | −0.84 | −0.32 | −0.27 | −0.02 | −0.15 | −0.92 | −1.02 | −0.48 | 0.03 | −1.03 | -3.35 |
| 2017 | −1.68 | −0.48 | −0.61 | −0.93 | −0.12 | −0.78 | −0.56 | −0.82 | −0.65 | −1.94 | −1.27 | −1.22 |
| 2018 | −0.82 | −0.15 | 0.30 | 0.08 | 0.21 | −0.32 | −0.47 | −0.57 | −1.24 | −0.83 | −0.69 | −1.28 |

**Table A11.** Monthly maximums for sea surface height for the years 2014–2018.

| Year\Month | Jan | Feb | Mar | Apr | May | Jun | Jul | Aug | Sep | Oct | Nov | Dec |
|---|---|---|---|---|---|---|---|---|---|---|---|---|
| 2014 | 2.60 | 1.04 | 0.43 | 1.27 | 1.59 | 1.57 | 1.34 | 0.33 | 0.49 | 1.53 | 2.81 | 0.11 |
| 2015 | 0.21 | 0.23 | 0.70 | 1.21 | 0.69 | 0.78 | 0.08 | 1.20 | 0.33 | 1.65 | −0.03 | 0.27 |
| 2016 | 0.16 | 0.27 | 1.01 | 1.04 | 1.71 | 1.23 | 1.06 | 0.24 | 0.24 | 2.37 | 0.44 | 0.05 |
| 2017 | 0.05 | 1.10 | 1.06 | 1.19 | 1.70 | 0.60 | 0.60 | −0.05 | 0.30 | −0.49 | −0.14 | 0.06 |
| 2018 | 0.70 | 0.90 | 1.57 | 1.42 | 2.18 | 1.50 | 1.83 | 0.22 | −0.20 | −0.34 | 1.01 | −0.29 |

**Table A12.** Monthly standard deviations for sea surface height for the years 2014–2018.

| Year\Month | Jan | Feb | Mar | Apr | May | Jun | Jul | Aug | Sep | Oct | Nov | Dec |
|---|---|---|---|---|---|---|---|---|---|---|---|---|
| 2014 | 0.73 | 0.52 | 0.20 | 0.26 | 0.29 | 0.39 | 0.23 | 0.15 | 0.12 | 0.39 | 0.62 | 0.24 |
| 2015 | 0.21 | 0.15 | 0.15 | 0.49 | 0.20 | 0.25 | 0.20 | 0.24 | 0.11 | 0.31 | 0.22 | 0.24 |
| 2016 | 0.17 | 0.14 | 0.19 | 0.21 | 0.36 | 0.22 | 0.29 | 0.19 | 0.10 | 0.34 | 0.17 | 0.49 |
| 2017 | 0.23 | 0.31 | 0.24 | 0.31 | 0.36 | 0.19 | 0.14 | 0.11 | 0.15 | 0.20 | 0.20 | 0.17 |
| 2018 | 0.35 | 0.20 | 0.22 | 0.17 | 0.44 | 0.40 | 0.40 | 0.13 | 0.15 | 0.07 | 0.33 | 0.21 |

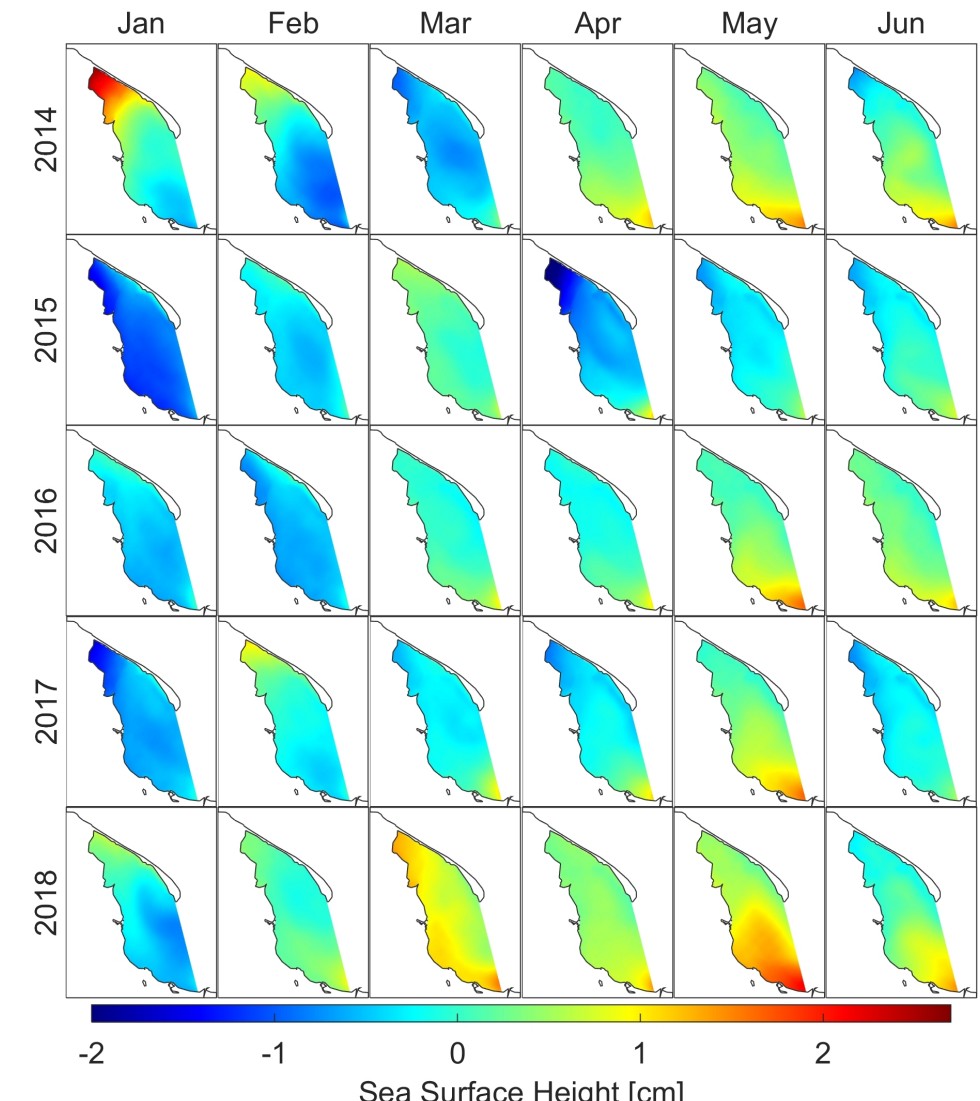

**Figure A19.** Monthly means of sea surface height for separated years. January to June.

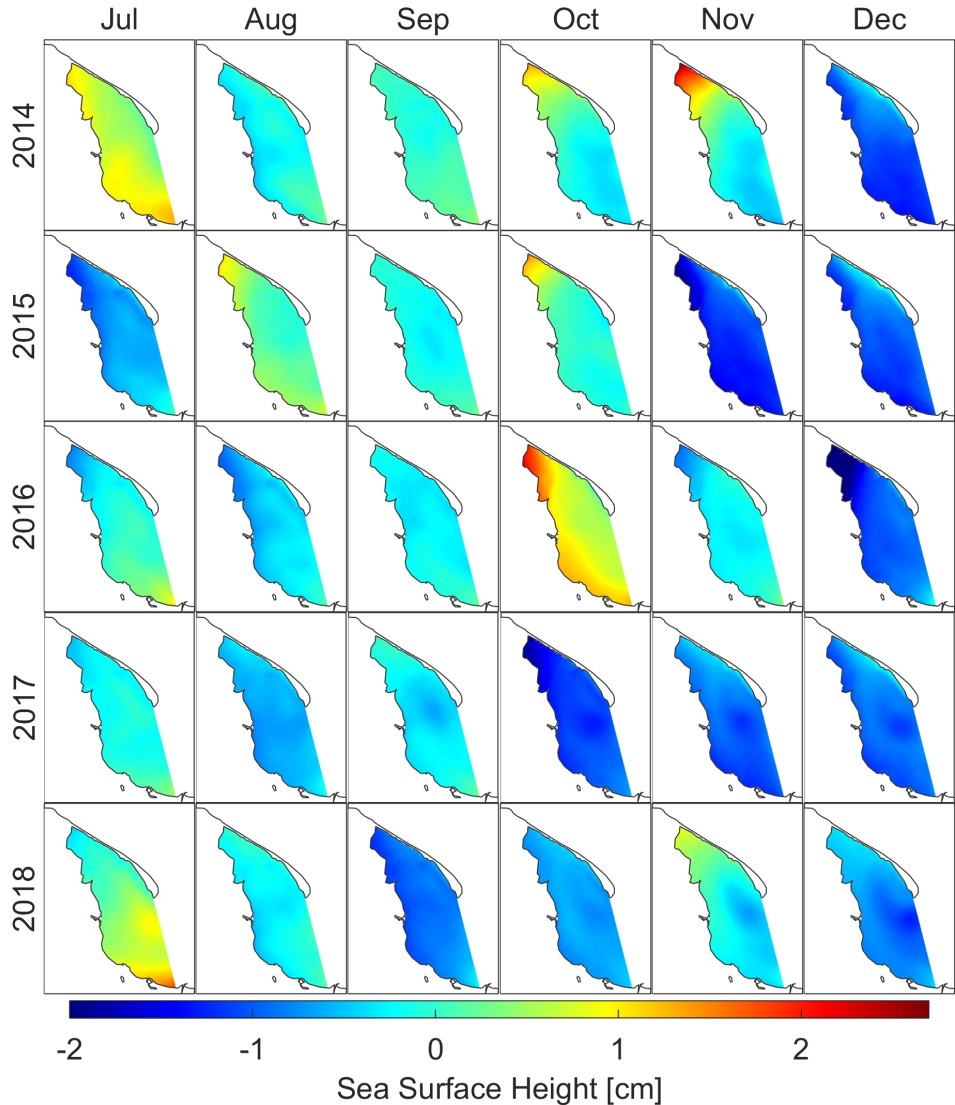

**Figure A20.** Monthly means of sea surface height for separated years. July to December.

*Appendix A.4. Currents*

**Table A13.** Monthly means for currents for the years 2014–2018.

| year\month | Jan | Feb | Mar | Apr | May | Jun | Jul | Aug | Sep | Oct | Nov | Dec |
|---|---|---|---|---|---|---|---|---|---|---|---|---|
| 2014 | 5.54 | 4.76 | 2.69 | 1.74 | 1.97 | 4.54 | 2.47 | 3.47 | 1.61 | 3.32 | 4.85 | 4.45 |
| 2015 | 4.95 | 3.43 | 2.07 | 5.31 | 2.81 | 3.66 | 3.96 | 2.48 | 1.57 | 2.98 | 4.98 | 5.27 |
| 2016 | 3.63 | 4.06 | 2.29 | 2.10 | 2.84 | 2.26 | 3.43 | 3.79 | 1.71 | 2.89 | 3.33 | 5.60 |
| 2017 | 4.53 | 3.24 | 3.41 | 3.57 | 2.57 | 3.13 | 2.55 | 2.25 | 2.26 | 3.90 | 3.85 | 4.64 |
| 2018 | 4.21 | 2.75 | 2.70 | 2.63 | 4.31 | 3.64 | 3.89 | 2.34 | 2.68 | 2.88 | 3.73 | 3.19 |
| mean | 4.57 | 3.65 | 2.63 | 3.07 | 2.90 | 3.45 | 3.26 | 2.87 | 1.97 | 3.19 | 4.15 | 4.63 |

**Table A14.** Monthly maximums for currents for the years 2014–2018.

| year\month | Jan | Feb | Mar | Apr | May | Jun | Jul | Aug | Sep | Oct | Nov | Dec |
|---|---|---|---|---|---|---|---|---|---|---|---|---|
| 2014 | 21.33 | 24.60 | 20.64 | 19.41 | 14.10 | 18.70 | 14.92 | 14.04 | 14.06 | 10.99 | 16.72 | 17.13 |
| 2015 | 20.94 | 23.54 | 21.32 | 19.88 | 12.44 | 15.64 | 17.05 | 13.81 | 10.90 | 13.08 | 17.67 | 16.91 |
| 2016 | 21.79 | 23.17 | 20.91 | 15.51 | 12.94 | 12.46 | 14.71 | 16.63 | 11.92 | 15.40 | 13.28 | 18.14 |
| 2017 | 20.89 | 22.95 | 20.47 | 13.98 | 14.99 | 15.94 | 16.33 | 9.00 | 9.95 | 12.92 | 13.51 | 16.13 |
| 2018 | 21.33 | 23.48 | 23.57 | 16.17 | 14.88 | 15.18 | 27.59 | 14.53 | 16.11 | 14.30 | 14.31 | 16.87 |

**Table A15.** Monthly standard deviations for currents for the years 2014–2018.

| year\month | Jan | Feb | Mar | Apr | May | Jun | Jul | Aug | Sep | Oct | Nov | Dec |
|---|---|---|---|---|---|---|---|---|---|---|---|---|
| 2014 | 3.55 | 2.63 | 1.70 | 1.47 | 1.45 | 3.40 | 1.73 | 2.04 | 1.19 | 1.79 | 3.18 | 1.68 |
| 2015 | 1.87 | 2.09 | 1.84 | 3.09 | 1.61 | 2.12 | 2.16 | 1.45 | 1.06 | 1.77 | 1.82 | 2.06 |
| 2016 | 1.96 | 1.66 | 1.97 | 1.67 | 1.75 | 1.66 | 2.55 | 2.17 | 1.31 | 1.63 | 1.36 | 2.64 |
| 2017 | 1.96 | 1.78 | 1.88 | 2.64 | 2.31 | 2.10 | 1.94 | 1.28 | 1.55 | 1.90 | 1.93 | 2.08 |
| 2018 | 2.08 | 2.31 | 2.50 | 1.89 | 3.11 | 2.55 | 3.59 | 1.53 | 1.51 | 1.48 | 2.43 | 1.84 |

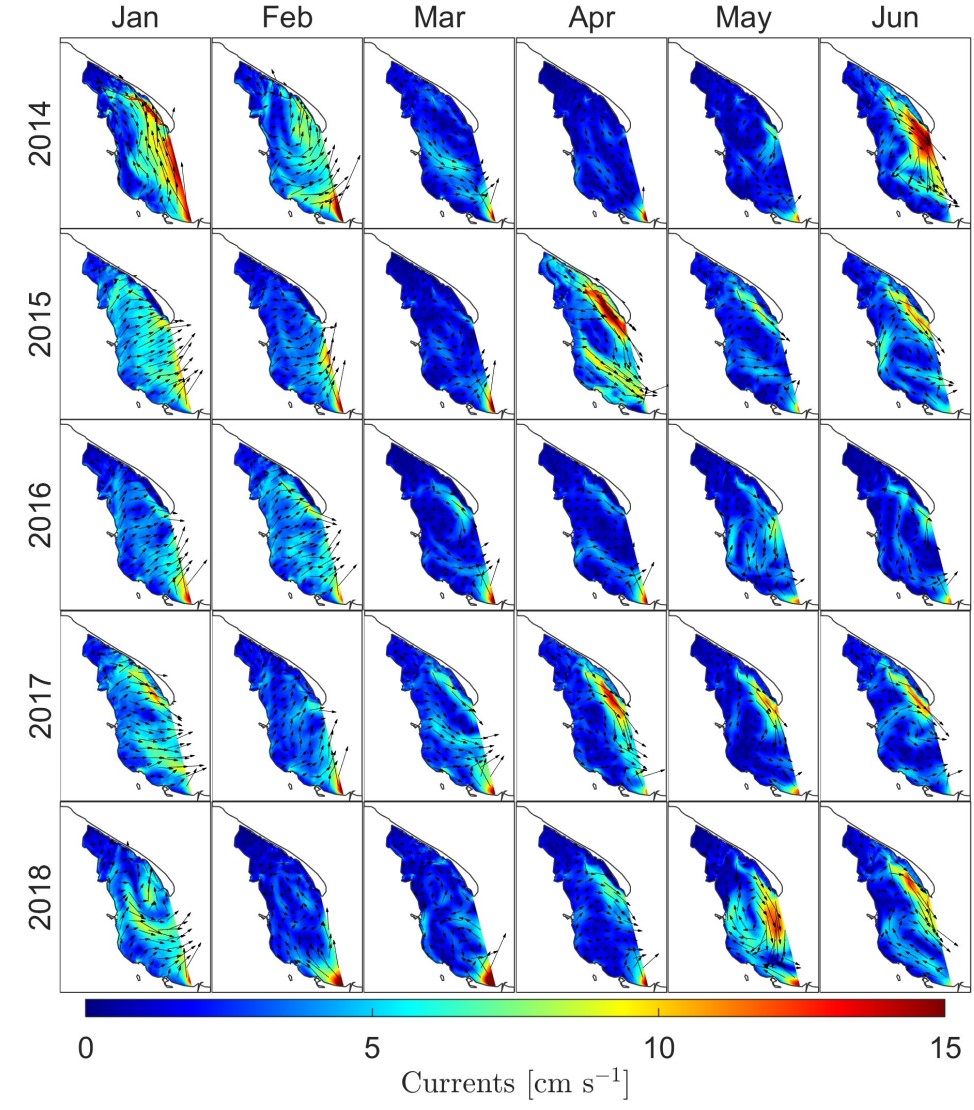

**Figure A21.** Monthly means of currents at surface level for separated years. January to June.

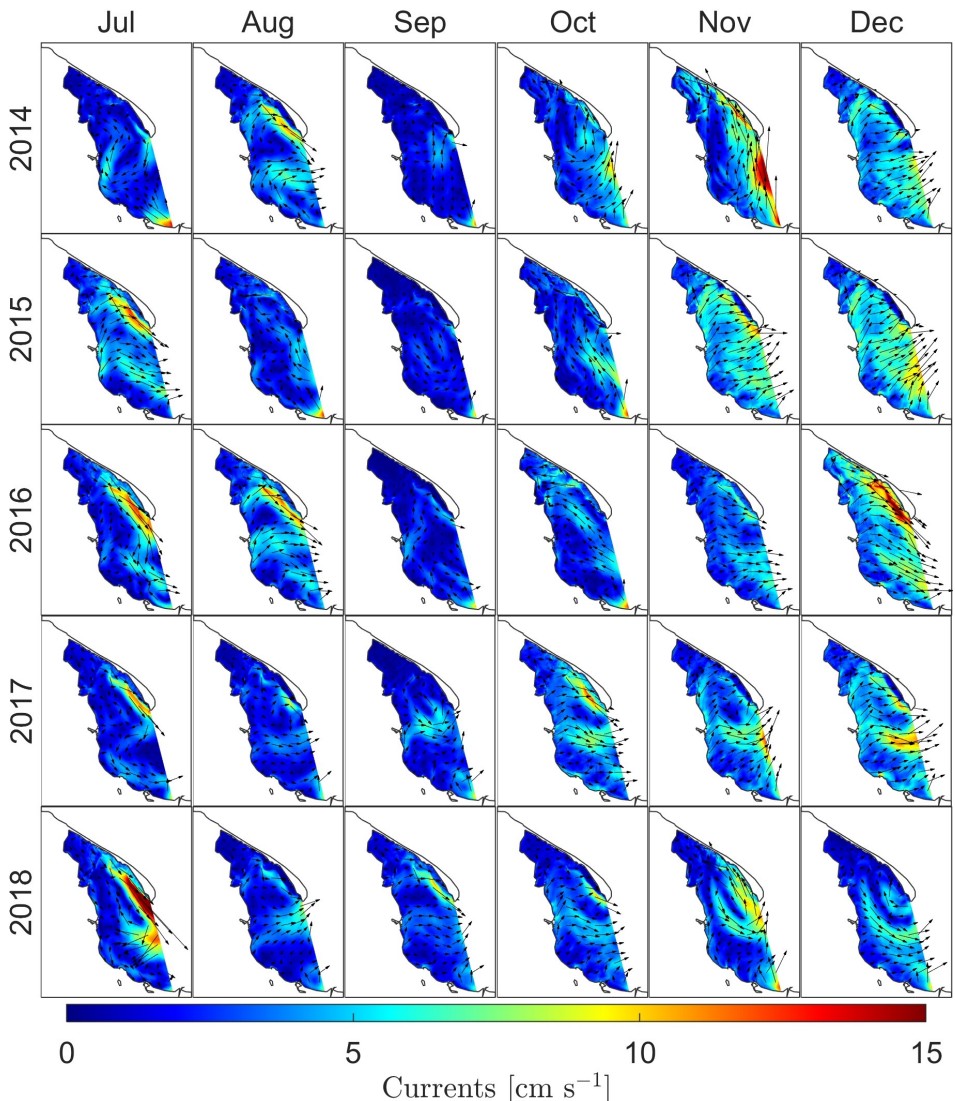

**Figure A22.** Monthly means of currents at surface level for separated years. July to December.

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
