# Peer review of "High-Resolution Ecosystem Model of the Puck Bay (Southern Baltic Sea)—Hydrodynamic Component Evaluation"

_water, doi:10.3390/w11102057_

Round 1

Reviewer 1 Report

Outline

The manuscript deals with the long term numerical analysis of Puck Bay. The “EcoPuckBay model system” is described and the main results of a 5 years analysis are presented and discussed. The topic is of interest for the Journal readers and I found the manuscript interesting and it is worth to be published. Nevertheless, I have some concerns and comments I ask the authors to address before to recommend the manuscript for publication. In the followings each concern is detailed.

Concerns

#1 – Model calibration

The major concern is related to the model calibration of the numerical system. Indeed, it is not clear to me if the authors performed a kind of calibration to achieve the results illustrated in section 3.1. Indeed, while I can understand that a series of parameters was selected to run the model (just for instance, zr, the roughness height described in Section 2.2, was selected as 0.5 cm), I do not catch if the value selection was performed by comparing the observed and computed results. I ask the authors to describe the model calibration, if applicable, or to discuss the model parameters selection, more in general.

#2 – Model forcing

My second (and last) major concern about the work is related to the model forcing. Section 2.4 describes the atmospheric forcing and section 2.5 the estimation of river discharge. I have a series of requests:

#2.1 – It seems that the tidal oscillations are not considered in the study. I guess they have been assessed as negligible in the case at hand. However, it should be discussed that the bay dynamic is not affected by tidal propagation. I guess that the problem is hidden in 3D CEMBS used as forcing field at the open boundary (section 2.3). However, I found this aspect not clearly discussed. The authors may refer to Medvedev et al. (2013).

#2.2 – Section 2.5 deals with the estimation of river discharge. It is not clear to me why the model Hype will be replaced by the model SWAT. In other words, it is not clear if the river discharge estimation is reliable enough and if the use of the model Hype (instead of Swat) affects the results. It seems that Hype model was used along with historical time series spanning the period 1980-2010 and the river discharges “have been calculated as a long-term means from the available 30-year period”. I understand that the precipitation information (section 2.4) was not used to estimate the river discharge and average discharge (monthly?) was used. Am I right? What about intense events? Why the model SWAT will be used only for river mouths located within the Puck District? Indeed, when discussing the salinity distribution, the manuscript states “This situation can be improved by attaching SWAT model data to the EcoPuckBay model.” (lines 262-263). In the concluding remarks, the manuscript states that the river discharge was estimated by means of hydrological model SWAT (line 458). In conclusion, I found the discussion about river discharge confusing.  This is important as some concluding remarks refer to the effects of river flow, e.g. “Fresh water inflow from rivers (mainly from the Vistula River) has the greatest influence on the variability of surface salinity values in the Puck Bay.” (lines 388-389). I ask the authors to clarify these aspects.

#2.3 – Section 2.3 discusses the open boundary conditions. A diagram showing the large-scale grid and the local grid may be of great help for the readers.

#3 – Web portal

Section 3.3 illustrates the web portal for data sharing. However, I found the section technical assistance to use it (a kind of user manual). I suggest restating the whole section by underlining the importance of information dissemination. The authors may refer to the paper by Lisi et al. (2019) that deals with “Management of Monitoring Data and Information Flow”

#4 – Conclusions

The section should start with a description of the study (i.e. aims, approaches, methods) and then should illustrate the results of the study.

#5 – Minor concerns

5.1 – Line 83. “the range of 7-8 with a deviation of around 1”. unit of measurement is missing.

5.2 – I suggest, if possible, to merge Figures 1, 2 and 3 (at least 1 and 2).

5.3 – Page 4 (after line 101, the line number is missing), check the sentence “The ocean component is EcoPuckBay is based on POP”

5.4 – Line 102. I suggest changing “The equation of hydrostatics:” with “The momentum equation along the vertical direction within the hydrostatic approximation:”

5.5 – Equation 4, a parenthesis is missing

5.6 – Line 123. “the volume data of river discharge come from the The” should be “the volume data of river discharge come from The”

5.7 – Line 210. “In the Figure 5 we present vertical water temperature” should be “In Figure 5 we present the average vertical water temperature”

5.8 – Figures 5 and 7. I suggest replacing “Model level” with “Normalized vertical elevation”, i.e. the local elevation normalized with water depth.

5.9 – Section 3.2.4. It could be interesting to discuss the results in terms of “currents” in relation to the results in terms of “sea surface height”.

5.10 – Line 337. “In the following article” should be “In this paper”

5.11 – Line 364. “experimental data” should be “field data”.

5.12 – Line 425. “Enforcing forces” should be “considered forces”.

Cited references

Medvedev, I. P., Rabinovich, A. B., & Kulikov, E. A. (2013). Tidal oscillations in the Baltic Sea. Oceanology, 53(5), 526-538.

Lisi, I., Feola, A., Bruschi, A., Pedroncini, A., Pasquali, D., & Di Risio, M. (2019). Mathematical Modeling Framework of Physical Effects Induced by Sediments Handling Operations in Marine and Coastal Areas. Journal of Marine Science and Engineering, 7(5), 149.

Reviewer 2 Report

This paper introduces a new numerical model named the EcoPuckMay model which was developed in order to evaluate marine environment in the Puck Bay, Poland. The model was validated by comparison with the observation, and showed better accuracy than the existing model. The simulation results were satisfyingly described in detail, however, the discussion looks insufficient. In summary, the research can be published after employment of some revisions.

Introduction:

(1) ll.41-52: The authors referred several models before introducing the EcoPuckBay model. However, the differences between the EcoPuckBay and the existing models are unclear for the reviewer. What is the advantage of the EcoPuckBay model?

Materials and methods:

(2) ll.106-108: The authors describes that the EcoPuckBay model has a spin-up region covering the "effective area". Please show the whole region including spin-up region in a figure.

Results:

(3) Although the EcoPuckBay model generally shows a better performance than NOMO-Nordic, there are some difference between the EcoPuckBay model calculation and the observation, especially in a vertical profiles (Fig.A7, A16). It might to be caused by that the model failed to accurately reproduce the thermal stratification. Adding some discussion about this would make the paper more novel. For example, what do you think is the main reason? Vertical eddy viscosity? Boundary conditions? Or grid resolution?

Others:

(4) There are some trivial errors.

l.45: ... an Integrated Information and Predictive Service for ... -> ... an integrated information and predictive service for ...

Eq.4: A closing bracket is missed.(The third term on the left side.)

The sentence following Eq.7: ... [lambda] latitude and longitude, [alpha] effective Earth radius ... -> ... [lambda], [phi] latitude and longitude, [alpha] effective Earth radius ...

l.123: ... come from the The ... -> ... come from the ...

ll.464-465: This will lower model uncertainties ... -> This will provide lower model uncertainties ...

Round 2

Reviewer 1 Report

The authors have addressed all of my concerns. I thank them for this. I found the manuscript improved. Nevertheless, I would have liked to read some of the satisfactory replies directly in the manuscript (for instance response 2.2 and response 3 to the other reviewer's concern).

However, aside all of that, it is my opinion that the manuscript can be accepted for publication in the Journal.